# Transformer Modeling for Both Scalability and Performance in Multivariate Time Series

## Abstract

Variable count is among the main scalability bottlenecks for transformer modeling in multivariate time series (MTS) data. On top of this, a growing consensus in the field points to indiscriminate inter-variable mixing as a potential source of noise-accumulation and performance degradation. This is likely exacerbated by sparsity of informative signals characteristic of many MTS systems coupled with representational misalignment stemming from indiscriminate information mixing between (heterogeneous) variables. While scalability and performance are often seen as competing interests in transformer design, we show that both can be improved simultaneously in MTS by strategically constraining the representational capacity of inter-variable mixing. Our proposed method, transformer with **Del**egate **T**oken **A**ttention (**DELTAformer**), constrains inter-variable modeling through what we call delegate tokens which are then used to perform full, unconstrained, inter-temporal modeling. Delegate tokens act as an implicit regularizer that forces the model to be highly selective about what inter-variable information is allowed to propagate through the network. Our results show that DELTAformer scales linearly with variable-count while actually outperforming standard transformers, achieving state-of-the-art performance across benchmarks and baselines. In addition, DELTAformer can focus on relevant signals better than standard transformers in noisy MTS environments and overall exhibit superior noise-resilience. Overall, results across various experiments confirm that by aligning our model design to leverage domain-specific challenges in MTS to our advantage, DELTAformer can simultaneously achieve linear scaling while actually improving its performance against standard, quadratic transformers.

## 1 Introduction

Accurate and reliable multivariate time series (MTS) forecasting is crucial across domains such as traffic flow modeling for smart city designs (Jin et al., 2022; Zhang et al., 2018), weather forecasts (Cheng & Yu, 2024; M et al., 2024), infectious disease forecasts (Mossop & Rahman, 2023; Said et al., 2020), and more. While past works have shown that even simple linear models could outperform complex models in MTS forecasting at the time (Zeng et al., 2023), recent approaches have adequately addressed this concern through variate-wise tokenization and inter-variable attention mechanism (Nie et al., 2023; Zhang & Yan, 2023), opening the door to powerful state-of-the-art transformer models in recent years (Liu et al., 2023; 2025; Zhou et al., 2021). **Note**: we use *variable* and *channel* inter-changeably in this work.

Despite their demonstrated capabilities, these transformers typically scale quadratically along the number of variables (Lan et al., 2025; Liu et al., 2023), or quadratic in both the number of variables and token (patch) positions (Bai & Liu, 2025; Liu et al., 2025; Nguyen et al., 2024). We denote these architectures *variate-only transformers* and *full transformers*, respectively (refer to Section B.2 for more detailed explanations).

We take variable count as the primary scaling concern for transformer design as it fundamentally limits model applicability. While sampling techniques offer workarounds for high-dimensional datasets, models incapable of scaling efficiently with variable count remain fundamentally constrained in both performance and computational efficiency.

In addition, recent works in the language domain have shown that transformers in practice tend to only allocate small proportions of attention scores to relevant information while disproportionately focusing on irrelevant context, leading to under-utilization of resources and accumulation of noise in representation learning (Leviathan et al., 2025; Ye et al., 2025). These findings may be even more relevant in MTS data where informative signals are often sparse (Lu et al., 2024; Rubin-Falcone et al., 2023; Zhao & Shen, 2024), and noise from unexpected temporal fluctuations and inter-variable heterogeneity can be significant (Huang et al., 2023; Zhou et al., 2024). In fact, a growing consensus in the field points to indiscriminate inter-variable mixing as a potential source of significant noise accumulation and performance degradation. Some works propose more sophisticated forms of inter-variable mixing (Zhao & Shen, 2024; Chen et al., 2024) while others simply disable inter-variable mixing entirely, opting for a channel-independent (CI) strategy instead (Chen et al., 2025; Nie et al., 2023).

By recognizing that inter-variable modeling is a source of both scaling bottleneck and noise accumulation in attention mechanism, we propose transformer with **Del**egate **T**oken **A**ttention (**DELTAformer**) to leverage and take advantage of these domain-specific challenges in MTS. DELTAformer constrains inter-variable modeling to achieve both scalability and noise-reduction simultaneously through what we call delegate tokens. DELTAformer first assigns a delegate token to represent patches across all variables at each patch position as an attention-based aggregation. Next, full-attention is performed among all delegate tokens, allowing for unconstrained inter-temporal interactions between constrained inter-variable representations. Finally, the delegate tokens that now contain both inter-variate and inter-temporal information are propagated back out to their respective patches in an attention-based propagation. By constraining the representation pathway of inter-variable information through delegate tokens, the model is forced to be highly selective about what inter-variable information it propagates through the network. This constraint may reduce the accumulation of spurious correlations, as reduced representation capacity would penalize uninformative data interactions more than full transformers. Overall, DELTAformer shows the ability to act as an implicit regularizer that can enhance signal detection in data with sparse information patterns while remaining robust to noise. DELTAformer scales linearly with the number of variables while achieving state-of-the-art performance on well-acknowledged benchmarks.

## 2 Preliminaries: Notation and Common Practices

**Multivariate Time Series Forecasting**  In MTS forecasting, the goal is to predict future time-steps $T_{t:t+\tau}$, $\tau \geq 1$ given historical samples $T_{1:t-1}$. The MTS data typically takes the shape $T \in \mathbb{R}^{C \times L}$, where $C$ denotes the variable (channel) dimension and $L$ denotes the look-back window.

**Variate-wise Patching**  It has been demonstrated and widely acknowledged that variate-wise patching (tokenization) is superior to point-wise patching or even time-wise patching (Nie et al., 2023; Zhang & Yan, 2023). By taking a collection of time-steps of length $P$ along the look-back window $L$ in each variable and tokenizing it, variate-wise patching preserves variable-specific information in each token which is critical for robust MTS modeling (Liu et al., 2023). This also results in overall data of shape $\mathbb{R}^{C \times d_{patch} \times L/P}$ where $d_{patch}$ is the patch dimension, which can trivialize the computational commitments along the time dimension $L$. DELTAformer takes full advantage of this by being quadratic along $L/P$ while remaining linear along the number of variables $C$.

**Variate-only and full transformers**  Here, we define two types of standard transformers that have seen success in MTS modeling in recent years: *full* and *variate-only* transformers. Full transformers (Bai & Liu, 2025; Liu et al., 2025; Nguyen et al., 2024) take as input MTS data tokenized with standard variate-wise patching. This results in $C \times (L/P)$ total tokens and allows each token at each variable and patch position to interact with every other token in a standard attention mechanism. The complexity of full transformers are typically $O(C^2(L/P)^2)$. Variate-only transformers (Lan et al., 2025; Liu et al., 2023; Wang et al., 2025) take variate-wise patching to the extreme and tokenize the entire look-back window $L$ for each variable, creating overall data of shape $\mathbb{R}^{C \times d_{patch}}$. Mathematically, this can be achieved by setting $P$ to be equal to $L$. While they have been shown to be highly performant, variate-only transformers still incur quadratic complexity with respect to variable count at $O(C^2)$.

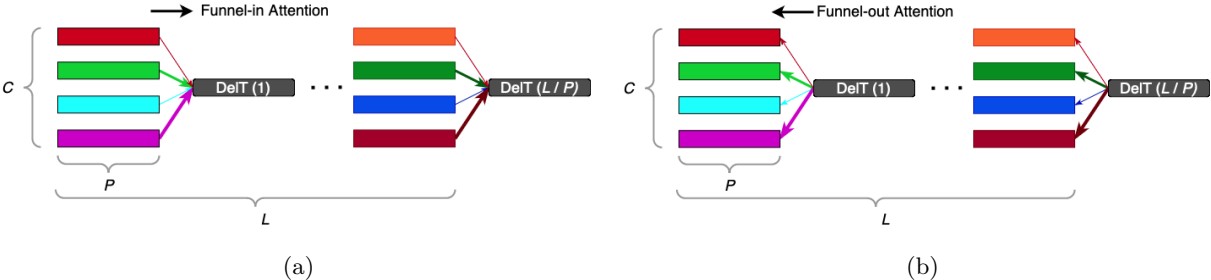

(a)                                     (b)

Figure 1: (a) Funnel-in attention: for each patch position, patches are funneled into a corresponding delegate token (DelT) in a weighted aggregation using the attention mechanism. The attention mechanism facilitates how much of each variable's heterogenous information contributes to the delegate token. (b) Funnel-out attention: After delegate token attention has been performed and each delegate token contains both inter-variate and temporal information, each delegate token funnels out its contents back to the corresponding patches. Funnel-out attention explicitly propagates information back to each patch, which allows the attention mechanism to learn to preserve variable-specific information and characteristics.

For more details on the different patching strategies and transformer architectures used in MTS modeling, refer to Section B.

## 3   Multivariate Time Series Forecasting with DELTAformer

Here we introduce DELTAformer, a scalable, efficient, and performant transformer model for MTS forecasting. DELTAformer is made up of three stages: funnel-in phase, Delegate Token Attention phase, and funnel-out phase.

### 3.1   Model Architecture

**Funnel-in Attention**   All patches at each patch position (of which there are $L/P$ patch positions) are assigned a learnable delegate token. Funnel-in attention is performed for each patch position where the attention mechanism facilitates how much contribution each variable-wise patch has on its delegate token. Formally, given patch positions $M$ and delegate tokens $D$ where $M \in \mathbb{R}^{C \times d_{patch} \times (L/P)}$ and $D \in \mathbb{R}^{(L/P) \times d_{patch}}$, funnel-in attention and overall delegate token update at $i^{\text{th}}$ patch position can be described as follows:

$$D_i^{funnel\_in} = softmax(\frac{D_i M_{[:,:,i]}^T}{\sqrt{d_M}})M_{[:,:,i]} \tag{1}$$

$$D_i' = LayerNorm(D_i^{funnel\_in} + MLP(D_i^{funnel\_in})) \tag{2}$$

where $D_i \in \mathbb{R}^{d_{patch}}$, $M_{[:,:,i]} \in \mathbb{R}^{C \times d_{patch}}$,   $\forall i \in [0 : L/P]$ and $D' \in \mathbb{R}^{d_{patch}}$ representing the updated delegate tokens. In practice, delegate tokens can be configured to be of arbitrary size $d'_{patch}$ as a multiple of $d_{patch}$ (called expansion factor). In such a case, $M$ is linearly projected to match the size $d'_{patch}$ before funnel-in attention.

Funnel-in attention for each $M_i$ and $D_i$ achieves $O(C)$ complexity. Over all patch positions, funnel-in attention's total complexity can be characterized as $O(C \times (L/P))$ where it typically holds that $L/P \ll C$. Visualization for funnel-in attention is shown in Figure 1a.

**Delegate Token Attention**   As a result of funnel-in attention, each delegate token is conditioned on inter-variable dependencies at the corresponding patch position, without temporal interactions. To introduce temporal modeling, delegate token attention is performed between all delegate tokens. Delegate token

attention is a standard self-attention mechanism allowing for pair-wise interactions as shown in Figure 2. This allows delegate tokens to learn even distant temporal dependencies, should they exist. Mathematically, delegate token attention takes the form,

$$\widehat{D}^{delta} = softmax(\frac{D'D'^T}{\sqrt{d_{D'}}})D' \tag{3}$$

$$D^{delta} = LayerNorm(\widehat{D}^{delta} + MLP(\widehat{D}^{delta})) \tag{4}$$

where $D^{delta} \in \mathbb{R}^{(L/P) \times d_{patch}}$. The overall complexity of delegate token attention is $O((L/P)^2)$

By taking advantage of the fact that $L/P$ is often much smaller than $C$, delegate token attention can be made a full attention mechanism without incurring a large memory footprint. In addition, because this attention mechanism is performed on delegate tokens and not individual patches, the risk of accumulating noise from spurious temporal correlations is reduced as the receptive field is made smaller and the model is forced to be more selective during funnel-in phase.

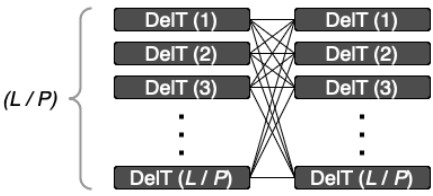

Figure 2: Delegate Token Attention for full pair-wise interactions between delegate tokens (DelT).

**Funnel-out Attention**   Funnel-out attention takes the fully conditioned delegate tokens $D^{delta}$ and propagates them back out to their original patches as illustrated in Figure 1b. The information being propagated back out at this stage contains both inter-variate and temporal information. Mathematically, funnel-out attention is defined as

$$M'_{[:,:,i]} = softmax(\frac{M_{[:,:,i]}D_i^{delta^T}}{\sqrt{d_{D^{delta}}}})D_i^{delta} \tag{5}$$

$$M^{final} = LayerNorm(M'_{[:,:,i]} + MLP(M'_{[:,:,i]})) \tag{6}$$

which results in the updated patch representations $M_{[:,:,i]}^{final} \in \mathbb{R}^{C \times d_{patch}}$, bringing the overall data back to shape $M^{final} \in \mathbb{R}^{C \times d_{patch} \times (L/P)}$. Funnel-out attention results in the same complexity as funnel-in attention.

Because funnel-out attention propagates information to each patch explicitly, it is able to learn to preserve variable-specific information such as data magnitudes and statistical distribution for each patch. As a result, each updated variable representation is able to reasonably maintain its unique characteristics despite information mixing that invariably happens during funnel-in and delegate token attention operations.

Overall, funnel-in attention, delegate token attention, and funnel-out attention make up a complete layer of DELTAformer. In practice, we find that conditioning the patches in a univariate fashion before the transformer layer helps with performance.

### 3.2   Moving Beyond Static Computational Pathways with Delegate Tokens

There are aspects of DELTAformer's architecture that mirror existing designs in MTS and beyond. Namely, funnel-in and funnel-out operations used for populating and propagating information through delegate tokens bear close similarities to works like Crossformer (Zhang & Yan, 2023) in MTS and more broadly PerceiverIO (Jaegle et al., 2022) and their use of routers and latent arrays, respectively. Likewise, these concepts can also be motivated as instances of virtual nodes (Qian et al., 2024; Shirzad et al., 2023) from graph neural networks (GNNs) (Hamilton et al., 2017; Gilmer et al., 2017). The typical unifying element in these works is their use of routers/latent arrays/virtual nodes as *computational pathways* to achieve efficiency in memory and sometimes information propagation (in the case of GNNs).

In contrast, DELTAformer goes beyond using its delegate tokens as mere computational pathways and uses these constrained representations directly in delegate token attention for temporal modeling. This represents the key distinction of DELTAformer's use of delegate tokens, forcing the model to learn selective aggregation and propagation strategies that preserve only the most predictive cross-variable relationships needed for temporal modeling. Thus, delegate tokens function not as computational pathways but as *constrained data representations* that can learn to capture sparse but meaningful patterns in MTS data. DELTAformer's superior ability to focus on relevant signals while remaining robust to noise empirically supports our claim that delegate tokens' constrained representations act as an implicit regularizer that discourages noise accumulation. Furthermore, DELTAformer's average performance improvement of 41% against Crossformer (Zhang & Yan, 2023) which only uses routers as computational pathways empirically highlights the efficacy of our approach. These results and additional analysis are fully explored next, in Section 4.

## 4 Experiments

We conduct extensive experiments to verify DELTAformer's robustness and capabilities. We start with controlled scenarios using synthetic data and artifical noise to analyze DELTAformer's attention allocation efficiency and robustness to noise compared to standard transformers. To demonstrate DELTAformer's viability in real world data across domains, we conduct extensive experiments against well-acknowledged benchmarks and compare DELTAformer's performance against recent state-of-the-art baseline models. In addition, we perform model analysis and ablation studies for DELTAformer and its components.

### 4.1 Attention Allocation Efficiency and Robustness to Noise

In this section, we evaluate DELTAformer's efficiency in attention allocation as well as its robustness in performance against datasets when noise is injected into them systematically.

**Attention Allocation Efficiency**  Recent works in foundational transformer research suggest that standard attention mechanisms tend to over-allocate attention scores to irrelevant context and use key-retrieval tasks to illustrate this point (Leviathan et al., 2025; Ye et al., 2025). This indicates that attention mechanisms are prone to inefficiencies, using only small proportions of available resources for informative modeling.

Motivated by these findings in the language domain, we introduce an experiment analogous to key-retrieval tasks to analyze the attention allocation efficiencies of transformers in MTS forecasting. To simulate key-retrieval objectives, we generate synthetic data and sparsely embed them with ground-truth signals as keys while keeping the rest of the data highly noisy and without discernible patterns. We use sine waves as the ground-truth signals and measure how much attention scores different transformer architectures allocate to the keys. In addition, we fix the number of keys and systematically increase the size of the overall dataset to measure the transformers' ability to remain focused on the keys against irrelevant context growing in both overall size and in proportion to the keys. For more details on the experimental setup and its limitations, refer to Section C.2.

As illustrated in Figure 3, DELTAformer is able to consistently dedicate higher proportions of its attention scores to the correct answers with growing dataset size. A standard variate-only transformer and full transformer dedicate around 30% less attention scores (weights) to the correct answers. In addition, DELTAformer shows the most resilience against growing size of irrelevant context by only losing around 14% of its attention scores on the keys overall. On the other hand, variate-only transformer and full transformer both lose around 50% of the attention scores dedicated to the keys over the course of growing dataset size.

These results indicate that standard transformers also exhibit attention allocation inefficiencies in MTS forecasting as they do in language modeling. Despite this, we find that this does not necessarily translate to degradation in forecasting accuracy. In fact, it is a testament to the robustness of standard transformers that they can maintain accurate forecasts despite under-utilizing their resources. However, DELTAformer's superior ability in attention allocation efficiency partly justifies our hypothesis that a fully quadratic transformer with indiscriminate pair-wise interactions are not always necessary in MTS. We further validate this in Section 4.2 by comparing DELTAformer's performance against state-of-the-art-models with real-world data.

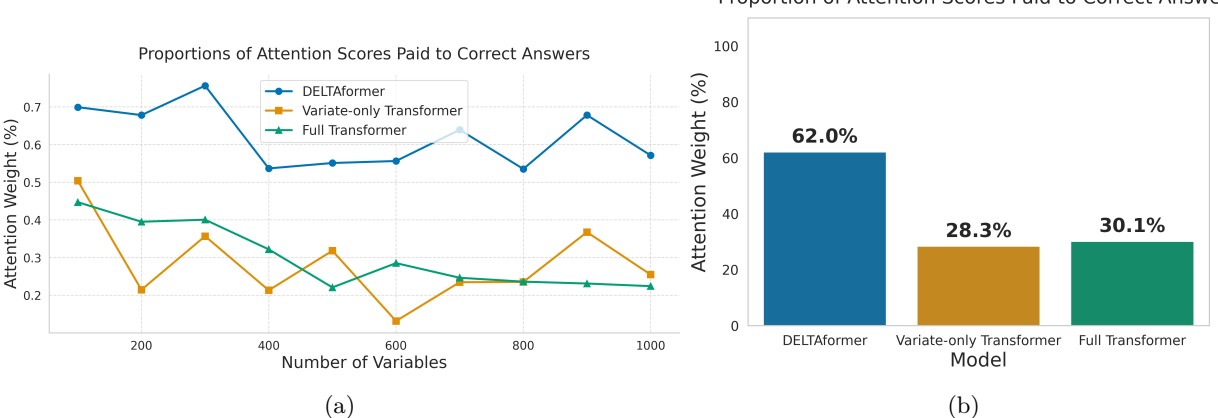

(a)                                  (b)

Figure 3: (a) plots the proportion of attention weights assigned to correct answers (keys) against a growing size of irrelevant context in a setup analogous to key-retrieval tasks in language modeling. DELTAformer (in blue) is not only able to focus higher proportions of attention scores to the keys but it also shows resilience with respect to growing size of irrelevant context. Variate-based transformer (in yellow) and full transformer (in green) generally assign around 30% less attention coefficients to the keys while also showing poor ability to focus on them with growing size and proportion of irrelevant context. (b) shows a histogram view. Overall, DELTAformer learns to dedicate twice as much attention weights to the correct answers compared to standard transformers.

**DELTAformer's Robustness to Noise**  We also analyze the performance profile of DELTAformer when the given data are exposed to varying degrees of noise, from low to high. The motivation for this experiment is simple. Unlike other data modalities, MTS data are particularly prone to noise and corruption. As such, we introduce this experiment as a systematic and repeatable way to measure models' performances against varying levels of noise.

We use three widely acknowledged benchmark datasets, ECL, Solar, and Traffic and inject artificial noise. We use recent state-of-the-art transformer models to compare against DELTAformer: iTransformer (Liu et al., 2023) and Timer-XL (Liu et al., 2025) as instances of variate-only and full transformer, respectively. We chart the growth of forecasting errors in all three models starting from baseline (no noise) and systematically increase the proportion of data affected by noise as illustrated in Figure 4 (lower values indicate better performance). For more details on the experimental setup and its limitations, refer to Section C.3.

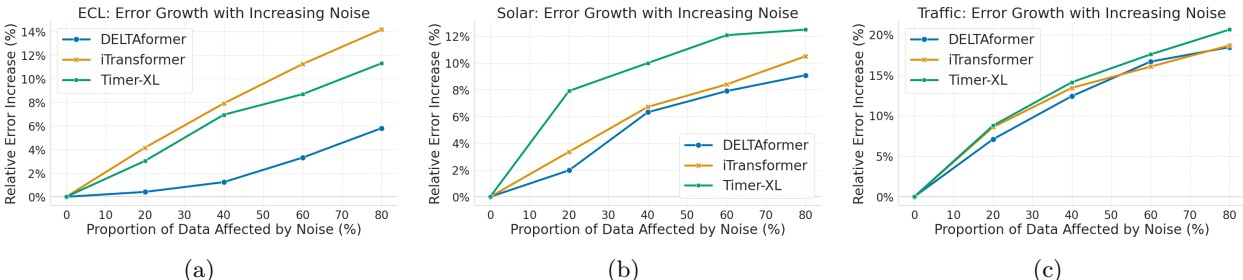

(a)                          (b)                         (c)

Figure 4: Error growth in forecasting accuracy versus proportion of data affected by noise with DELTAformer (in blue), iTransformer (Liu et al., 2023) (in yellow), and Timer-XL (Liu et al., 2025) (in green) across 3 datasets: (a) ECL, (b) Solar, (c) Traffic. Overall, DELTAformer generally shows superior ability to discern noise and make accurate predictions even at higher levels of noise in the data compared to iTransformer and Timer-XL. Lower values indicate better performance.

In all three datasets, DELTAformer generally shows superior resilience against noise present in the data across various noise levels. DELTAformer shows particularly strong performance with the ECL dataset, only

exhibiting around 6% in forecasting error growth from baseline while iTransformer and Timer-XL show 12% and 14% growth in forecasting errors, respectively.

These findings, while informative, stem from synthetic datasets with sparse signals or artificially injected noise. These experimental designs may inherently favor constrained architectures. Real-world time series data on the other hand, contain complex, non-stationary patterns where the signal versus noise distinctions are rarely clear-cut. As such, we perform extensive experiments on well-acknowledged benchmarks against state-of-the-art models to further validate our model's performance in the next section. We find that DELTAformer remains robust against various real-world data across domains, achieving state-of-the-art results in many cases.

## 4.2 Main Forecasting Results

In this section, we provide the main forecasting results on 12 benchmarks including 8 long-term and 4 short-term prediction datasets. Long-term benchmarks include ECL, ETT (4 subsets), Traffic, and Weather datasets from (Wu et al., 2021) along with Solar dataset from (Lai et al., 2018). For short-term benchmarks, we use 4 datasets from PEMS (Guo et al., 2022). More information on the benchmark datasets, model configuration, and experimental setup are in Section A.

**Baselines** We carefully select 10 recent state of the art models to compare against our work including (a) transformer-based methods: Timer-XL (Liu et al., 2025), SimpleTM (Chen et al., 2025), iTransformer (Liu et al., 2023), Crossformer (Zhang & Yan, 2023), PatchTST (Nie et al., 2023), Autoformer (Wu et al., 2021); (b) MLP-based method: TimerMixer (Wang et al., 2024); (c) Graph Neural Network based method: CrossGNN (Huang et al., 2023); and (d) linear model: DLinear (Zeng et al., 2023). For transformer models, we constrain pre-processing steps to be simple and widely-used techniques (patching, RevIN) and isolate the transformer components from models that make use of complex data-engineering and data-enriching techniques to ensure fair comparison.

Table 1: Forecasting results (96 time-steps as look-back window length): We make extensive comparisons against state-of-the-art models under various forecasting horizons (prediction lengths) following the experimental setup used in Chen et al. (2025); Liu et al. (2023). Forecast horizons are {12, 24, 48, 96} time-steps for PEMS and {96, 192, 336, 720} time-steps for the rest. Reported results are averaged across the forecast horizons with full results available in Section E. The best result in each dataset is marked in **red** while the second best result in each dataset is underlined and marked in blue.

| Models | DELTAformer (Ours) | | Timer-XL (2025) | | SimpleTM (2025) | | TimeMixer (2024) | | iTransformer (2024) | | CrossGNN (2023) | | Crossformer (2023) | | PatchTST (2023) | | DLinear (2023) | | Autoformer (2021) | |
|---|---|---|---|---|---|---|---|---|---|---|---|---|---|---|---|---|---|---|---|---|---|
| Metrics | MSE | MAE | MSE | MAE | MSE | MAE | MSE | MAE | MSE | MAE | MSE | MAE | MSE | MAE | MSE | MAE | MSE | MAE | MSE | MAE |
| ETT (Avg.) | **0.360** | **0.391** | 0.391 | 0.402 | _0.364_ | _0.388_ | 0.376 | 0.394 | 0.383 | 0.399 | 0.376 | 0.395 | 0.685 | 0.578 | 0.381 | 0.397 | 0.442 | 0.444 | 0.458 | 0.459 |
| ECL | **0.165** | **0.263** | _0.173_ | _0.266_ | 0.204 | 0.290 | 0.182 | 0.272 | 0.178 | 0.270 | 0.201 | 0.300 | 0.244 | 0.334 | 0.205 | 0.290 | 0.212 | 0.300 | 0.227 | 0.338 |
| Traffic | **0.418** | **0.277** | 0.442 | 0.291 | 0.509 | 0.323 | 0.497 | 0.300 | _0.428_ | _0.282_ | 0.583 | 0.323 | 0.550 | 0.304 | 0.481 | 0.304 | 0.625 | 0.383 | 0.628 | 0.379 |
| Weather | **0.244** | **0.275** | 0.295 | 0.323 | 0.264 | 0.285 | _0.245_ | _0.276_ | 0.258 | 0.278 | 0.247 | 0.289 | 0.259 | 0.315 | 0.259 | 0.281 | 0.265 | 0.317 | 0.338 | 0.382 |
| Solar-Energy | _0.229_ | **0.257** | 0.309 | 0.310 | **0.210** | 0.267 | 0.237 | 0.290 | 0.233 | _0.262_ | 0.249 | 0.313 | 0.641 | 0.639 | 0.270 | 0.307 | 0.330 | 0.401 | 0.885 | 0.711 |
| PEMS (Avg.) | _0.116_ | 0.223 | 0.187 | 0.267 | 0.208 | 0.295 | **0.091** | **0.198** | 0.119 | _0.218_ | – | – | 0.220 | 0.304 | 0.217 | 0.306 | 0.320 | 0.394 | 0.615 | 0.575 |

**Results** Forecasting results are detailed in Table 1. The forecasting accuracy are reported in both Mean Squared Error (MSE) and Mean Absolute Error (MAE) where lower values indicate better performance.

DELTAformer achieves state-of-the-art performance on all of the long-term forecasting datasets. On high-dimensional datasets such as Traffic and ECL, DELTAformer achieves 2.3% and 4.6% reductions in MSE compared to previous state-of-the art models, respectively. On the low-dimensional datasets such as ETTm2 and ETTh2, DELTAformer shows 18.3% and 4.5% MSE reductions, respectively, compared to previous state-of-the art models. Among patch-based transformers, DELTAformer's superior performance is more pronounced, achieving an average MSE reduction of 41% and 16% against Crossformer and PatchTST, respectively.

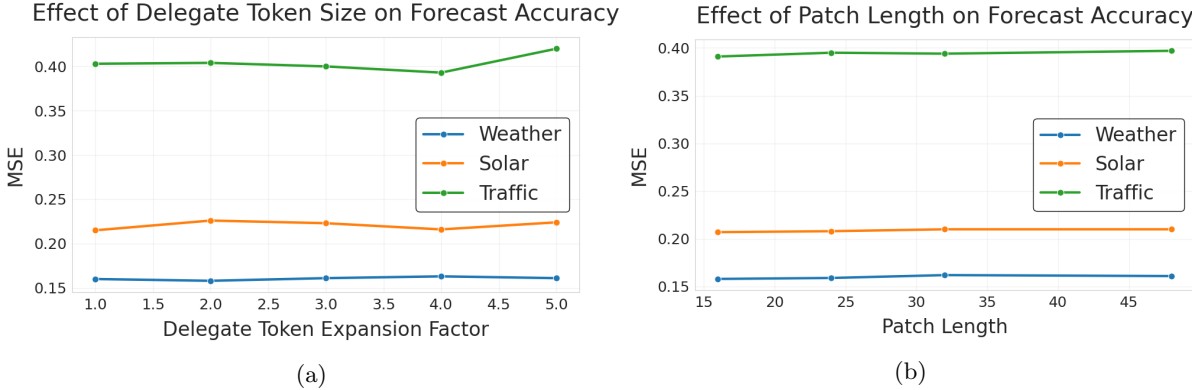

(a)
(b)

Figure 5: (a) Effect the size of delegate tokens have on forecasting performance across Weather, Solar, and Traffic datasets. Expansion factor refers to expansion proportion of delegate token from the original patch dimension size (ex. expansion factor of 2.0 indicates that the delegate token dimension will be twice as large as patch dimension) (b) Effect various patch length has on forecasting performance across Weather, Solar, and Traffic datasets. Overall, we observe that our model is generally robust to various patch lengths.

On the short-term forecasting benchmarks, we observe that transformers in general do not show the same robust performance as they do in long-term benchmarks. Instead, it is TimeMixer (Wang et al., 2024), an MLP-based architecture that dominates in short-term forecasting. TimeMixer uses multiscale mixing optimizations to capture immediate temporal patterns with seasonal and trend components. This may help short-term forecasting where sudden changes can dominate performance, highlighting the utility of advanced pre-processing techniques for MTS forecasting though transformer models (including ours) typically keep pre-processing simple for benchmarking purposes. In addition, DELTAformer and other transformers all model temporally distant dependencies as part of their attention mechanism. While this can be crucial for long-term forecasts, it may hinder their abilities to isolate and model short-term interactions.

### 4.3 Model Analysis

**Funneling Operation Analysis**   We analyze the significance of attention mechanism used during funneling phases of DELTAformer by systematically replacing the attention mechanism with MLP and linear layers. The results are shown in Table 2. Overall, we find that attention mechanism's unique ability to explicitly aggregate and propagate variable-specific information is crucial for robust MTS modeling, achieving up to 26% and 27% improvements in MSE reduction compared to MLP and linear-layer based funneling, respectively. More details are discussed in Section C.4.

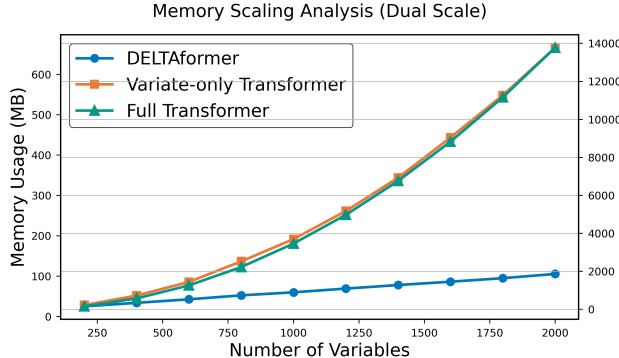

Figure 6: DELTAformer (in blue) shows linear scaling against variable count (x-axis) compared to standard transformers (in yellow and green) which show quadratic growth.

**Ablation Studies**   We conduct ablation studies to analyze the effects the size of each delegate token and overall patch length have on forecasting performance. Intuitively, bigger the size of delegate tokens, the less bottleneck it will impose on in-coming and out-going information. We find that DELTAformer is generally immune to large performance shifts with the changing size of delegate tokens. This empirically confirms that information patterns in MTS data tend to be sparse and that increasing the size of delegate tokens does not necessarily result in more accurate and robust modeling.

Table 2: Comparing DELTAformer's performance with different funnel-in and funnel-out methods. We replace the attention mechanism with MLP and linear models to study the effects that different funneling operations have on overall representation learning and forecasting ability. Overall, we find that attention mechanism outperforms the other methods, particularly in high dimensional datasets.

| Datasets | Traffic | | ECL | | Weather | |
|---|---|---|---|---|---|---|
| Metrics | MSE | MAE | MSE | MAE | MSE | MAE |
| Funnel-in **Attention**, Funnel-out **Attention** | **0.388** | **0.266** | **0.137** | **0.236** | 0.159 | **0.205** |
| Funnel-in **Attention**, Funnel-out **MLP** | 0.481 | 0.306 | 0.146 | 0.248 | **0.157** | 0.206 |
| Funnel-in **MLP**, Funnel-out **Attention** | 0.431 | 0.297 | 0.138 | 0.236 | 0.163 | 0.212 |
| Funnel-in **MLP**, Funnel-out **MLP** | 0.525 | 0.339 | 0.160 | 0.260 | 0.164 | 0.212 |
| Funnel-in **Linear**, Funnel-out **Linear** | 0.534 | 0.338 | 0.157 | 0.258 | 0.160 | 0.208 |
| Time-only Transformer | 0.647 | 0.357 | 0.260 | 0.358 | 0.395 | 0.427 |

As for patch-length ablation, we find that DELTAformer is generally robust and stable across various patch lengths, which would indicate that large proportions of DELTAformer's performance comes from its constrained inter-variable modeling. This shows that temporal dependencies may be more information-dense and easily capturable than inter-variable dependencies. This is in line with findings from (Liu et al., 2023). Additional details on ablation studies can be found in Section D.

**Model Scaling**  To visually illustrate DELTAformer's scalability, its memory usage at various dataset sizes are charted along with standard transformer architectures (variate-based and full) in Figure 6. As it can be seen, DELTAformer clearly shows linear profile in memory growth while standard transformers grow exponentially as the number of variables increase. More analysis on DELTAformer's scalability and additional memory analysis on DELTAformer and other transformer baselines are found in Section C.1.

## 5  Related Work

**Multivariate Time Series Forecasting**  Early deep learning approaches relied on recurrent architectures to model temporal dependencies (Lai et al., 2018; Miller et al., 2024; Shih et al., 2019). However, RNNs have shown to suffer from gradient stability issues particularly for long sequences. Temporal Convolutional Networks (TCNs) addressed some limitations of RNNs using dilated convolutions for long-range dependencies. SCINet (LIU et al., 2022) introduced sample convolution and interaction blocks with multi-resolution analysis, achieving competitive results on benchmarks. Graph-based approaches (Huang et al., 2023; Li et al., 2017; Liu et al., 2022a) explicitly model inter-variable relationships and have been shown to be particularly effective on domains with existing spatial inductive bias such as traffic networks.

The success of transformers in language modeling (Vaswani et al., 2017) inspired their adaptations in time series forecasting (Wu et al., 2021; Zhou et al., 2021; 2022). However, subpar tokenization and pre-processing strategies led to surprising findings demonstrating that even simple linear models could outperform transformers in MTS (Toner & Darlow, 2024; Zeng et al., 2023). This has been sufficiently addressed in recent years through channel-independent (variate-wise) tokenization (Nie et al., 2023), revealing that preserving variable-specific information in each token was crucial for performance. Since then, MTS forecasting has seen a wave of transformer models achieving state-of-the-art performance in various benchmarks (Chen et al., 2025; Liu et al., 2023; 2025; Wang et al., 2025).

In a parallel but equally important direction, recent research have also focused on advanced pre-processing and data-enriching strategies for robust MTS forecasting (Chen et al., 2023; Ekambaram et al., 2023; Wang et al., 2024; Zhong et al., 2023). These approaches keep the core model simple (usually an MLP) to showcase their sophisticated pre-processing and information mixing techniques. Transformer models (including DELTAformer) typically take the opposite approach and keep pre-processing simple to measure

core model performance. These two research directions aim to solve different, but related problems, working toward a common goal of robust MTS modeling.

## 6 Conclusion

We introduced DELTAformer, a transformer-based architecture for MTS forecasting that achieves both scalability and performance simultaneously through delegate token attention mechanism. By strategically aggregating information through delegate tokens, DELTAformer effectively captures both temporal dynamics and inter-variable correlations while reducing memory requirements and noise accumulation compared to standard transformers that are quadratic with respect to variable count. Extensive experiments demonstrate DELTAformer's state-of-the-art performance on standard benchmarks as well as its superior attention allocation efficiency and robustness to noise. Together, these properties make DELTAformer a viable and practical alternative to standard transformers in MTS. Limitations of DELTAformer and future directions are discussed in Section C.5.

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

## A  Implementation Details

### A.1  Dataset Information

We provide additional information on the real-world datasets used in our experiments. ECL, ETT (4 variations), Traffic, and Weather datasets are from (Wu et al., 2021). Solar dataset is from (Lai et al., 2018) and PEMS (4 variations) datasets were introduced in (Guo et al., 2022) and used for MTS forecasting in (LIU et al., 2022). We use dataset information and descriptions from (Liu et al., 2023; Wang et al., 2024; Zhang & Yan, 2023).

1. **ETTh1 & ETTh2** contain 7 indicators of an electricity transformer in two years, including oil temperature, load characteristics, etc. Data points are recorded every hour.

2. **ETTm1 & ETTm2** contain the same data as with the ETTh datasets but recorded over 15 minute intervals.

3. **Weather** contains 21 meteorological indicators in the U.S over the entire year of 2020, including features like visibility, wind speed, etc.

4. **Solar** records solar power production of 137 PV plants in 2006, sampled every 10 minutes.

5. **ECL** records the hourly electricity consumption data of 321 clients.

6. **Traffic** records road occupancy rates measured by 862 sensors in San Francisco freeways from 2015 to 2016.

7. **PEMS (03, 04, 07, 08)** contains traffic flow data in California at 5 minutes intervals.

Table 3: Details of the datasets used in our experiments. The ETT datasets used training, validation, and testing split of 6:2:2 while the rest of the datasets used a 7:1:2 splits, in line with the experimental setups from (LIU et al., 2022; Liu et al., 2023; Zhang & Yan, 2023) and other baselines.

| Dataset | Dim | Prediction Length | Dataset Size | Frequency | Information |
|---|---|---|---|---|---|
| ETTh1 | 7 | {96, 192, 336, 720} | (8545, 2881, 2881) | Hourly | Electricity |
| ETTm1 | 7 | {96, 192, 336, 720} | (34465, 11521, 11521) | 15min | Electricity |
| ETTh2 | 7 | {96, 192, 336, 720} | (8545, 2881, 2881) | Hourly | Electricity |
| ETTm2 | 7 | {96, 192, 336, 720} | (34465, 11521, 11521) | 15min | Electricity |
| Weather | 21 | {96, 192, 336, 720} | (36792, 5271, 10540) | 10min | Weather |
| ECL | 321 | {96, 192, 336, 720} | (18317, 2633, 5261) | Hourly | Electricity |
| Traffic | 862 | {96, 192, 336, 720} | (12185, 1757, 3509) | Hourly | Transportation |
| Solar-Energy | 137 | {96, 192, 336, 720} | (36601, 5161, 10417) | 10min | Energy |
| PEMS03 | 358 | {12, 24, 48, 96} | (15617, 5135, 5135) | 5min | Transportation |
| PEMS04 | 307 | {12, 24, 48, 96} | (10172, 3375, 3375) | 5min | Transportation |
| PEMS07 | 883 | {12, 24, 48, 96} | (16911, 5622, 5622) | 5min | Transportation |
| PEMS08 | 170 | {12, 24, 48, 96} | (10690, 3548, 3548) | 5min | Transportation |

## A.2 Metrics Used

For metrics used in our benchmarking, we utilize both mean squared error (MSE) and mean absolute error (MAE). MSE penalizes larger errors more heavily due to its quadratic nature, making it particularly sensitive to outliers. This characteristic is valuable for applications where large deviations are especially problematic. Conversely, MAE treats all error magnitudes linearly, providing a more robust measure of average model performance that is less influenced by occasional extreme predictions. By reporting both metrics, we offer a more comprehensive evaluation of model performance: MSE highlights models that avoid significant errors, while MAE better reflects the typical prediction accuracy a user might expect in practice.

$$MSE(y, \hat{y}) = \frac{1}{n} \sum_{i=1}^{n} (y_i - \hat{y}i)^2 \tag{7}$$

$$MAE(y, \hat{y}) = \frac{1}{n} \sum i = 1^n |y_i - \hat{y}_i| \tag{8}$$

## A.3 Experimental Details

All experiments are implemented in PyTorch (Paszke, 2019) and run on a single NVIDIA A100 80GB GPU. Forecasting results are run 3 times and averaged. We use the ADAM (Kingma & Ba, 2014) optimizer and MSE loss function for all training. For more information on dataset-specific model configuration, refer to Table 4. For fair comparison against other transformer models, we follow the practices of recent state-of-the-art transformer baselines (Liu et al., 2023; 2025; Nie et al., 2023; Zhang & Yan, 2023) and prepare our data with simple pre-processing steps and prediction block. Concretely, this means that we only use a linear layer for tokenization and do not make any use of more complicated signal processing techniques or data enriching strategies to prime the model. For prediction, this also means that we are only using a linear layer as decoder for forecasting. These steps are taken strictly for benchmarking purposes and more sophisticated data engineering and forecasting methodologies should be integrated as seen fit in practical applications.

## A.4 Stability of Main Experimental Results

In this section, we provide the standard deviation of DELTAformer's forecasting performance across 3 independent runs, reported in Table 5. We report that DELTAformer's forecasting accuracy is stable and repeatable across datasets.

Table 4: Model configurations across different datasets

| Dataset | Model Hyper-parameter | | | | Training Process | | |
|---|---|---|---|---|---|---|---|
| | Expansion Factor[*] | Layers | Patch Length ($P$) | Learning Rate | Loss | Batch Size | Epochs |
| ETTm1 | 1.5 | 2 | 16 | $10^{-3}$ | MSE | 128 | 10 |
| ETTm2 | 1.5 | 2 | 16 | $10^{-3}$ | MSE | 128 | 10 |
| Weather | 1.0 | 2 | 16 | $10^{-4}$ | MSE | 128 | 10 |
| Electricity | 1.5 | 2 | 16 | $10^{-4}$ | MSE | 4 | 10 |
| Solar | 1.5 | 2 | 16 | $10^{-4}$ | MSE | 16 | 10 |
| Traffic | 1.5 | 2 | 16 | $10^{-4}$ | MSE | 4 | 10 |
| PEMS | 1.5 | 2 | 16 | $10^{-4}$ | MSE | 32 | 20 |

[*] Expansion Factor controls the size of the delegate tokens as a multiple of the patch dimension.

Table 5: Forecasting performance and their standard deviations after 3 independent runs.

| Horizon | Solar | | ECL | | Traffic | |
|---|---|---|---|---|---|---|
| | MSE | MAE | MSE | MAE | MSE | MAE |
| 96 | 0.199±0.003 | 0.243±0.003 | 0.137±0.000 | 0.236±0.001 | 0.388±0.001 | 0.266±0.001 |
| 192 | 0.228±0.005 | 0.256±0.003 | 0.155±0.000 | 0.252±0.000 | 0.408±0.000 | 0.270±0.000 |
| 336 | 0.251±0.005 | 0.277±0.003 | 0.174±0.001 | 0.272±0.001 | 0.426±0.002 | 0.278±0.001 |
| 720 | 0.255±0.000 | 0.279±0.000 | 0.195±0.001 | 0.293±0.002 | 0.449±0.000 | 0.292±0.000 |

| Horizon | Weather | | ETTm2 | | ETTh2 | |
|---|---|---|---|---|---|---|
| | MSE | MAE | MSE | MAE | MSE | MAE |
| 96 | 0.159±0.001 | 0.205±0.001 | 0.163±0.002 | 0.261±0.001 | 0.237±0.005 | 0.324±0.004 |
| 192 | 0.206±0.000 | 0.252±0.000 | 0.195±0.003 | 0.293±0.000 | 0.291±0.000 | 0.362±0.000 |
| 336 | 0.263±0.000 | 0.293±0.000 | 0.237±0.002 | 0.325±0.001 | 0.349±0.006 | 0.402±0.004 |
| 720 | 0.348±0.000 | 0.348±0.000 | 0.313±0.001 | 0.373±0.001 | 0.474±0.009 | 0.472±0.005 |

# B    Definitions and Background Information

## B.1    Patching Strategies

In this section, we go over the types of patching (tokenization) strategies employed for MTS data. There are 3 main types of patches: point-wise patching, time-wise patching, and variate-wise patching.

**Point-wise patching**    is the simplest strategy for patching MTS data. It simply takes each time-step as a token and optionally projects it to a suitable model dimension as illustrated in Figure 7a where each token is represented as a green dot. This was quickly determined by researchers to be subpar. While tokenizing the atomic unit of data is acceptable in some domains (such as language where each word or less represents a token), individual time-steps in MTS data contain too little information and thus cannot be used to model useful patterns. In addition, tokenizing each time step would result in $C \times L$ total tokens, which makes standard transformer modeling infeasible with even moderate-size datasets.

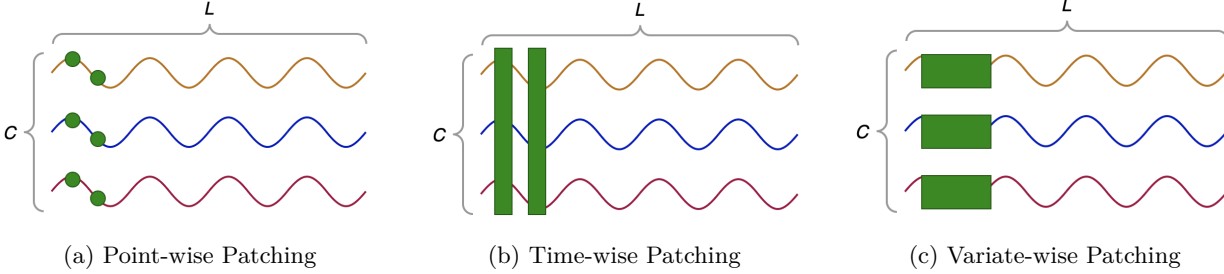

(a) Point-wise Patching        (b) Time-wise Patching        (c) Variate-wise Patching

Figure 7: Patching Strategies: (a) point-wise patching simply takes each time-step as a token and optionally projects it to a suitabl model dimension. Visual examples are shown as green dots. (b) time-wise patching is a patching strategy that tokenizes ach time-position made up of time-steps from all variables at that time position. Examples of time-wise patches are shown in vertical green patches. In reality, they would not overlap and cover the entire length of $L$. (c) variate-wise patching creates patches along each variable, creating $L/P$ tokens per variable. They are shown as green rectangles in the figure. Variate-wise patching is known to be the most robust of the 3 patching strategies and is used in our work.

**Time-wise patching** is a patching strategy that tokenizes each time-position which is made up of time-steps from all variables at the same time position. This process creates $L$ total tokens and is visualized in Figure 7b. While this was considered the standard way to tokenize MTS data at the time, it was later shown to be subpar even compared to simple linear models (Zeng et al., 2023). Time-wise patching strategy's inability to preserve variable-specific information was determined to be a major reason for this subpar performance (Nie et al., 2023; Liu et al., 2023). By simply taking information from all variables at each time-position and tokenizing it, each token is not able to maintain variable-independence or capture any multivariate correlations.

**Variate-wise patching** takes the opposite approach to time-wise patching and creates patches along each variable, creating $L/P$ tokens per variable, each of length $P$ as shown in Figure 7c. By tokenizing time-steps along each variable, variate-wise patches not only encode local temporal patterns but also preserve variable-specific characteristics, such as numerical magnitudes, distribution patterns, and periodicity that may be unique to each variable. Variate-wise patching has allowed recent transformers in MTS to achieve state-of-the-art performance across domains, sufficiently addressing concerns of transformer viability in MTS forecasting (Nie et al., 2023; Zeng et al., 2023).

## B.2 Variate-only and full transformers

In our work, we frequently address two types of standard transformer architectures that have seen popularity in recent years: variate-only and full transformers. Here, we define the two transformer architectures. We note that both variate-only and full transformers employ variate-wise patching, as detailed in Section B.1.

Full transformers (Bai & Liu, 2025; Liu et al., 2025; Nguyen et al., 2024) take as input MTS data tokenized with standard variate-wise patching as illustrated in Figure 7c. This results in $C \times (L/P)$ total tokens and allows each token at each variable and patch position to interact with every other token in a standard attention mechanism. Recent approaches have shown their utility particularly in modeling inter-variable and temporal dependencies in a unified fashion. However, this architectural pattern typically incurs complexity of $O(C^2(L/P))$, which hinders their scalability. However, with practical and GPU-specific optimizations, this approach has shown to be practically more efficient than their theoretical complexity (Liu et al., 2025).

Variate-only transformers (Lan et al., 2025; Liu et al., 2023) take variate-wise patching to the extreme and tokenize the entire look-back window $L$ for each variable. This creates $C$ total tokens where each token represents a variable's entire input temporal sequence and allows for attention mechanism that performs inter-variable modeling without explicit temporal interactions. This approach has shown to be performant in recent years and is generally more memory-efficient than full transformers. However, variate-only transformers still incur the quadratic bottleneck with respect to the variable-count at $O(C^2)$.

Both variate-only and full transformers have been demonstrably more capable and robust than transformers utilizing time-wise patching (Wu et al., 2021; Zhou et al., 2021; 2022) (Figure 7b). These architectures have achieved state-of-the-art performance across domains in recent years.

### B.3 Time-only Transformers

While funneling operations share some semantic similarities to time-wise patching used in time-only transformers, we observe two critical design decisions that fundamentally distinguish DELTAformer from time-only transformers. First, DELTAformer uses variate-wise patching as opposed to time-wise patching, allowing for preservation of variable-specific information as noted in (Liu et al., 2023; Nie et al., 2023; Zhang & Yan, 2023). This allows DELTAformer to outperform time-only transformer even when using linear layers for funneling. Second, using attention mechanism for funneling creates direct connections between each patch and its delegate token, allowing for more explicit modeling. During funnel-in phase, this means delegate tokens can control how much contribution to receive from each patch. During funnel-out phase, this means delegate tokens can explicitly propagate information that is relevant to each patch, preserving variable-specific characteristics.

### B.4 Data Non-stationarity

MTS data exhibits pronounced non-stationarity characterized by persistent alterations in statistical attributes and joint distributions across time (Liu et al., 2022b). To tackle this issue, reverse instance normalization (RevIN) (Kim et al., 2022) has been widely used in recent works. RevIN addresses distribution shift in MTS data by first normalizing input data using instance-specific statistics and then de-normalizing the model output to restore the original distribution. More formally,

$$\text{Normalization:} \quad \hat{x} = \gamma \cdot \frac{x - \mu_x}{\sigma_x} + \beta \tag{9}$$

$$\text{Denormalization:} \quad \hat{y} = \sigma_x \cdot \tilde{y} + \mu_x \tag{10}$$

where $\mu_x$ and $\sigma_x$ are the instance-specific mean and standard deviation, while $\gamma$ and $\beta$ are learnable parameters.

Variate-wise patching and RevIN make up two of the most widely used pre-processing techniques in transformers for MTS forecasting. For the most part, however, recent works keep pre-processing and data engineering simple and straight-forward to ensure fair and consistent comparisons against other works for benchmarking purposes. We follow this paradigm in our experiments.

## C Full Model Analysis

### C.1 Scalability Analysis

We provide additional details on scalability analysis for DELTAformer. In order to isolate the effects of the core transformer block, we implement variate-wise and full transformer to faithfully represent typical architectures seen in recent MTS forecasting. We forego any pre-processing or prediction blocks and focus only on the memory usage within the transformer block. We use 1 attention head and fix the number of time-steps. Then we run DELTAformer, variate-wise transformer, and full transformer on synthetic data and plot their memory usage across increasing variable count. We find that even with flash attention enabled, both variate-wise and full transformers do exhibit quadratic growth in memory against increasing variable count. On the other hand, DELTAformer shows a linear pattern against increasing variable count as expected from its complexity.

We show memory scaling up to 5000 variables in Figure 8 which also uses a dual scale with DELTAformer and variate-wise transformer using the left y-axis and full transformer using the right y-axis. While full transformer is an order of magnitude more memory inefficient than variate-wise transformer, they still exhibit a similar memory scaling pattern, visually showcasing their quadratic scaling. We also show a log-log plot

in Figure 9 to confirm DELTAformer's superior scalability. Finally, we provide peak memory allocation for DELTAformer along with iTransformer (Liu et al., 2023) and Timer-XL (Liu et al., 2025) in Table 7 against ECL, Traffic, and Weather to gauge practical implications for DELTAformer's linear scaling against variable count. We keep the batch size consistent for fair comparison and run their official code (refer to Section A.3 for experimental setup). For ECL, we find that DELTAformer achieves 41% and 93% reductions in peak memory allocation against iTransformer and Timer-XL, respectively. For Traffic which has close to 3 times more variables than ECL, DELTAformer achieves 80% and 97% reductions in peak memory allocation. For Weather, DELTAformer shows around 41% and 67% reductions in peak memory allocation.

Figure 8: We show memory usage of DELTAformer (in blue), variate-only transformer (in yellow), and full transformer (in green) against growing variable count up to 5000 variables. The plot uses a dual scale to account for the high magnitudes in memory usage of full transformer. DELTAformer and variate-only transformer use the left scale and full transformer uses the right. This plot clearly displays the linear growth in memory usage of DELTAformer while both variate-only and full transformers grow quadratically with increasing variable count.

Table 6: Complexity breakdown of various transformer designs

| Model Type | Complexity |
|---|---|
| Variate-wise Transformer (Lan et al., 2025; Liu et al., 2023) | $O(C^2)$ |
| Full Transformer (Bai & Liu, 2025; Liu et al., 2025; Nguyen et al., 2024) | $O(C^2(L/P)^2)$ |
| DELTAformer (ours) | $O((L/P)^2 + C(L/P))$ |

Table 7: Peak memory allocation of different models across multiple datasets (in MB)

| Model | ECL | Traffic | Weather |
|---|---|---|---|
| iTransformer (Liu et al., 2023) | 1194.83 | 5376.33 | 691.77 |
| Timer-XL (Liu et al., 2025) | 8106.52 | 43622.87 | 1248.84 |
| **DELTAformer** | **703.95** | **1098.92** | **405.40** |

## C.2 Attention Allocation Analysis

We create a synthetic data filled with random noise with fixed sequence length at 10,000 time-steps. Each look-back window length and prediction horizon are set at 100 time-steps. We start with 100 variables and

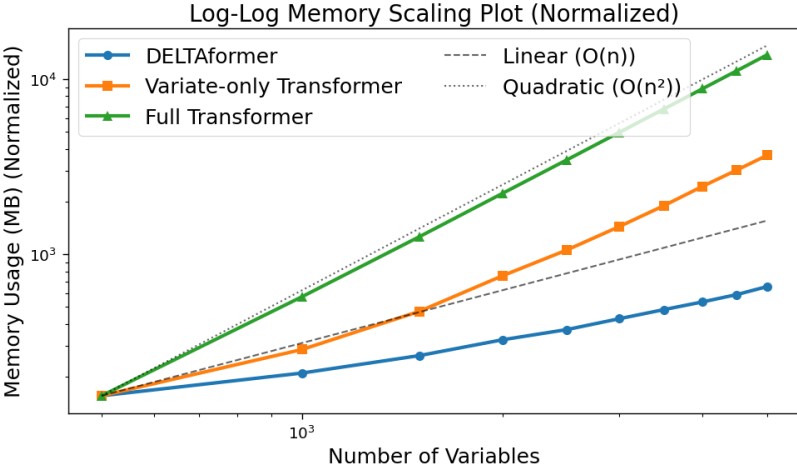

Figure 9: We show a log-log plot for DELTAformer (in blue), variate-only transformer (in yellow), and full transformer's memory usage against growing variable count up to 5000 variables. Log-log plots display data on a graph where both axes use logarithmic scales insteead of linear ones. They are particularly useful for visualizing power law relationships. We also include reference lines to show linear scaling and quadratic scaling. Full transformer shows quadratic scaling while variate-only transformer stays in between linear and quadratic under logarithmic scales. DELTAformer shows superior scalability compared to both variate-only and full transformers.

systematically increase their numbers in increments of 100, ending at 1000 variables. The number variables selected to be the keys are fixed at 10 and their features are replaced with ground-truth signals. As a result, with the growing number of variables from 100 to 1000, not only does irrelevant context grow in overall size, they also grow in proportion to the keys. We choose simple sine waves to be the ground-truth signals and populate the prediction targets with the ground truth signals, training the model to focus on the sparse keys present in an otherwise noisy and random data.

**Limitations** While this experiment provides valuable insights into attention allocation efficiencies of various models, there are some limitations that warrant consideration. First, the fixed-position sine waves used as keys create an idealized sparsity pattern that differs from real-world scenarios where relevant signals may be distributed or be intertwined with noise. Likewise, these artificial signal and noise patterns may inherently favor bottleneck architectures like DELTAformer, putting standard transformers at a disadvantage. As a result, findings from this experiment alone should not be used to make general claims about models' capabilities.

## C.3   Noise Robustness Analysis

For noise robustness analysis, artificial noise is injected at various levels along both variables and time-steps. The proportion of noise present in the data starts at 0% (baseline) and is increased in increments of 20%, ending at 80%. As an example, 20% noise means that 20% of both time positions and variables are affected by noise. We used Gaussian noise to be added to the affected data. While other types of noise patterns can also be used in place of Gaussian noise, we found that artifical injection of highly unpredictable noise (i.e. adding random spikes) tend to make the models untrainable at higher noise levels. Since these are real-world data with existing noise and various non-trivial patterns, we found Gaussian noise to be a good middle-ground that added complexity to the original datasets while preserving their fundamental characteristics.

**Limitations** This experiment has some limitations worth considering. First, the random Gaussian noise may fail to capture real-world and domain-specific noise patterns, potentially overestimating model robustness. Real-world noise may be much more unpredictable, sparse, and disruptive. In addition, we only introduce noise in a very structured manner, injecting noise into randomly selected time positions and variables with

fixed probabilities. While this was done intentionally to provide a systemic and repeatable experimental setting, it may not reflect noise patterns present in real-world data.

## C.4   Funneling Operation Analysis

We analyze the significance of the attention mechanism used during funneling phases of delegate token attention. As shown in Table 2, we systematically replace the attention mechanisms used in both funnel-in and funnel-out operations with MLP and linear layers against Traffic, ECL, and Weather datasets. We also include performance of a standard time-only transformer as funneling operations share some similarities to time-wise patching used in time-only transformers (refer to Section B.3 for more information on time-only transformers).

Overall, using the attention mechanism for both funnel-in and funnel-out operations achieved the best results as expected. In the cases where the attention mechanism was used only for one of the two funneling stages, we observe that using attention during funnel-out stage is more useful. We determine that attention mechanism's unique ability to dynamically weigh and propagate information back to each patch explicitly during funnel-out phase were critical in achieving robust performance and preserving variable-specific characteristics. Likewise, the subpar performance seen when using MLP for funnel-in phase indicate that attention mechanism's ability to dynamically adjust the amount of contribution of each patch plays an important role in conditioning delegate tokens effectively.

The effects of using less discerning models for funneling operations were more apparent in higher dimensional data such as Traffic and ECL. Weather datset on the other hand, with only 21 variables and mostly smooth and predictable data characteristics, was not affected significantly by the choice of funneling operations.

With respect to time-only transformers, we observe that DELTAformer outperforms them across datasets even when using linear layers for funneling operations. When using attention, DELTAformer achives a 40% MSE reduction in Traffic and 60% MSE reduction in Weather compared to a time-only transformer. This highlights the representation capability of variate-wise patching strategy compared to time-wise patching. Furthermore, its ability to explicitly aggregate and propagate variable-specific information when using attention mechanism for funneling fundamentally distinguishes DELTAformer from time-only transformers.

## C.5   Limitations and Future Directions

We discuss some of DELTAformer's limitations and potential future directions in this section. First, due to the architectural pattern of the funneling operations, DELTAformer may struggle in datasets with dense inter-variable information. While our assumption of sparse informative signals is observed to hold in many MTS systems, this can still put strain on DELTAformer's performance on some datasets. In such cases, it may be crucial to experiment with larger delegate token sizes to find the right balance. In addition, we motivate DELTAformer's design mainly through observations from prior works and empirical findings from our own experiments. While this was enough to design a performant and scalable model in this work, it may be useful in future work to develop more rigorous theoretical basis for DELTAformer's robust performance.

# D   Ablation Studies

We provide full ablation studies in this section. We use Solar, Weather, and Traffic datasets to measure performance.

On studying the effects that various delegate token sizes have on performance, we report a complicated and non-linear relationship. While performance is generally robust across different delegate token sizes, we do not see a clear trend between delegate token size and performance. Instead, we observe that as the size of delegate tokens increases, the performance becomes more unpredictable, potentially indicating that larger delegate tokens may be more sensitive to hyper-parameters and generally harder to train. On the other hand, this empirically confirms that information patterns in MTS data tend to be sparse and that increasing the size of delegate tokens does not necessarily result in more accurate and robust modeling.

With respect to various patch lengths, we observe that DELTAformer is generally immune to large performance shifts. However, across all three datasets, we do observe that patch length 16 for total input length of 96 results in the best performance. With increasing patch length, there also seems to be minor growth in forecasting error. This is in line with the model architecture as larger patch lengths would indicate less total tokens for delegate token attention, and thus modeling temporal dependencies become less robust with larger patch sizes.

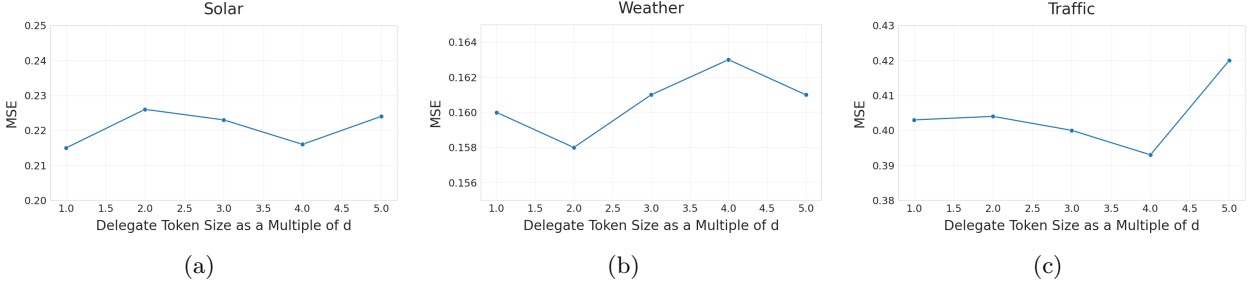

Figure 10: Plotting model performance against different delegate token dimension size as a multiple of the patch dimension.

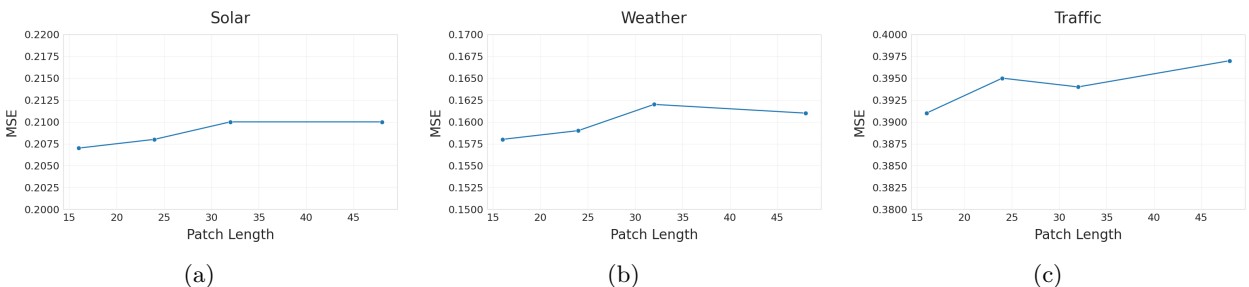

Figure 11: Plotting model performance against different patch lengths.

## E Full Results

**Long-term Forecasts** We provide the full results for long-term forecasting benchmarks in Table 8. Here, we report the forecasting errors in both MSE and MAE across each prediction length (96, 192, 336, 720) and also report their averages. The input length is fixed at 96 for all prediction lengths. In addition, we provide the complete performance details for the individual ETT datasets of which there are 4 in total.

We observe that DELTAformer achieves comparable or superior performance in 6 of the 8 total benchmarks while notably struggling against datasets of particularly small variable count, such as the ETTh1 dataset. Overall, we observe that transformer models in general such as iTransformer (Liu et al., 2023), SimpleTM (Chen et al., 2025), and Timer-XL (Liu et al., 2025) perform better than other architectures.

**Short-term Forecasts** We provide the complete results for short-term forecasting on 4 PEMS datasets in Table 9. We report the forecasting errors in both MSE and MAE across each prediction length (12, 24, 48, 96) with the input length fixed at 96 time-steps. We observe that TimeMixer (Wang et al., 2024), an MLP-based model outperforms all the other baselines including DELTAformer across the short-term forecasting benchmarks. We attribute TimeMixer's ability to process seasonal and trend based patterns as well as keeping with a simple model critical components in their dominant performance in short-term forecasting. Transformers in general struggle with short-term forecasts, including DELTAformer.

**Prediction versus Ground-Truth** We also chart predictions against grount-truth signals across Traffic, Solar, and ECL datasets to show visually how DELTAformer performs across diverse domains. For each

Table 8: Full long-term forecasting results (96 time-steps predicting {96, 192, 336, 720} time-steps): We make extensive comparisons against state-of-the-art models under various forecasting horizons (prediction lengths) following the experimental setup used in (Chen et al., 2025; Liu et al., 2023). The average performance across all forecast horizons for each dataset are also reported.

| Models | | DELTAformer (Ours) | | Timer-XL (2025) | | SimpleTM (2025) | | TimeMixer (2024) | | iTransformer (2024) | | CrossGNN (2023) | | Crossformer (2023) | | PatchTST (2023) | | DLinear (2023) | | Autoformer (2021) | |
|---|---|---|---|---|---|---|---|---|---|---|---|---|---|---|---|---|---|---|---|---|---|
| **Metrics** | | MSE | MAE | MSE | MAE | MSE | MAE | MSE | MAE | MSE | MAE | MSE | MAE | MSE | MAE | MSE | MAE | MSE | MAE | MSE | MAE |
| ETTm1 | 96 | **0.321** | **0.358** | 0.329 | 0.365 | 0.324 | 0.362 | 0.328 | 0.363 | 0.334 | 0.368 | 0.335 | 0.373 | 0.404 | 0.426 | 0.329 | 0.367 | 0.345 | 0.372 | 0.505 | 0.475 |
| ETTm1 | 192 | 0.373 | 0.388 | 0.393 | 0.403 | 0.367 | **0.383** | **0.364** | 0.384 | 0.377 | 0.391 | 0.372 | 0.390 | 0.450 | 0.451 | 0.367 | 0.385 | 0.380 | 0.389 | 0.553 | 0.496 |
| ETTm1 | 336 | 0.436 | 0.422 | 0.458 | 0.440 | 0.399 | 0.406 | **0.390** | **0.404** | 0.426 | 0.420 | 0.403 | 0.411 | 0.532 | 0.515 | 0.399 | 0.410 | 0.413 | 0.413 | 0.621 | 0.537 |
| ETTm1 | 720 | 0.525 | 0.473 | 0.581 | 0.501 | 0.463 | 0.441 | 0.458 | 0.445 | 0.491 | 0.459 | 0.461 | 0.442 | 0.666 | 0.589 | **0.454** | **0.439** | 0.474 | 0.453 | 0.671 | 0.561 |
| ETTm1 | Avg | 0.413 | 0.410 | 0.440 | 0.427 | 0.388 | **0.398** | **0.385** | 0.399 | 0.407 | 0.410 | 0.393 | 0.404 | 0.513 | 0.496 | 0.387 | 0.400 | 0.403 | 0.407 | 0.558 | 0.517 |
| ETTm2 | 96 | **0.163** | 0.261 | 0.179 | 0.262 | 0.178 | 0.260 | 0.176 | **0.259** | 0.180 | 0.264 | 0.176 | 0.266 | 0.287 | 0.366 | 0.175 | 0.259 | 0.193 | 0.292 | 0.255 | 0.339 |
| ETTm2 | 192 | **0.195** | **0.293** | 0.246 | 0.305 | 0.242 | 0.302 | 0.242 | 0.303 | 0.250 | 0.309 | 0.240 | 0.307 | 0.414 | 0.492 | 0.241 | 0.302 | 0.284 | 0.362 | 0.281 | 0.340 |
| ETTm2 | 336 | **0.237** | **0.325** | 0.313 | 0.348 | 0.305 | 0.344 | 0.304 | 0.342 | 0.311 | 0.348 | 0.304 | 0.345 | 0.597 | 0.542 | 0.305 | 0.343 | 0.369 | 0.427 | 0.339 | 0.372 |
| ETTm2 | 720 | **0.313** | **0.373** | 0.422 | 0.411 | 0.405 | 0.401 | 0.393 | 0.397 | 0.412 | 0.407 | 0.406 | 0.400 | 1.730 | 1.042 | 0.402 | 0.400 | 0.554 | 0.522 | 0.433 | 0.432 |
| ETTm2 | Avg | **0.227** | **0.313** | 0.290 | 0.332 | 0.283 | 0.327 | 0.278 | 0.325 | 0.288 | 0.332 | 0.282 | 0.330 | 0.757 | 0.610 | 0.281 | 0.326 | 0.350 | 0.401 | 0.327 | 0.371 |
| ETTh1 | 96 | 0.385 | 0.402 | 0.378 | 0.396 | **0.370** | **0.393** | 0.381 | 0.401 | 0.386 | 0.405 | 0.382 | 0.398 | 0.423 | 0.448 | 0.414 | 0.419 | 0.386 | 0.400 | 0.449 | 0.459 |
| ETTh1 | 192 | 0.465 | 0.447 | 0.428 | 0.426 | **0.424** | **0.422** | 0.440 | 0.433 | 0.441 | 0.436 | 0.427 | 0.425 | 0.471 | 0.474 | 0.460 | 0.445 | 0.437 | 0.432 | 0.500 | 0.482 |
| ETTh1 | 336 | 0.492 | 0.460 | 0.470 | 0.450 | **0.451** | 0.446 | 0.501 | 0.462 | 0.487 | 0.458 | 0.465 | **0.445** | 0.570 | 0.546 | 0.501 | 0.466 | 0.481 | 0.459 | 0.521 | 0.496 |
| ETTh1 | 720 | 0.511 | 0.486 | 0.497 | 0.475 | 0.480 | 0.477 | 0.501 | 0.482 | 0.503 | 0.491 | **0.472** | **0.468** | 0.653 | 0.621 | 0.500 | 0.488 | 0.519 | 0.516 | 0.514 | 0.512 |
| ETTh1 | Avg | 0.463 | 0.449 | 0.443 | 0.437 | **0.431** | 0.435 | 0.458 | 0.445 | 0.454 | 0.447 | 0.437 | **0.434** | 0.529 | 0.522 | 0.469 | 0.454 | 0.456 | 0.452 | 0.496 | 0.487 |
| ETTh2 | 96 | **0.237** | **0.324** | 0.296 | 0.347 | 0.285 | 0.344 | 0.292 | 0.343 | 0.297 | 0.349 | 0.309 | 0.359 | 0.745 | 0.584 | 0.302 | 0.348 | 0.333 | 0.387 | 0.346 | 0.388 |
| ETTh2 | 192 | **0.291** | **0.362** | 0.380 | 0.400 | 0.354 | 0.386 | 0.374 | 0.395 | 0.380 | 0.400 | 0.390 | 0.406 | 0.877 | 0.656 | 0.388 | 0.400 | 0.477 | 0.476 | 0.456 | 0.452 |
| ETTh2 | 336 | **0.349** | 0.402 | 0.430 | 0.437 | 0.365 | **0.399** | 0.428 | 0.433 | 0.428 | 0.432 | 0.426 | 0.444 | 1.043 | 0.731 | 0.426 | 0.433 | 0.594 | 0.541 | 0.482 | 0.486 |
| ETTh2 | 720 | 0.474 | 0.472 | 0.457 | 0.462 | **0.411** | **0.434** | 0.454 | 0.458 | 0.427 | 0.445 | 0.445 | 0.444 | 1.104 | 0.763 | 0.431 | 0.446 | 0.831 | 0.657 | 0.515 | 0.511 |
| ETTh2 | Avg | **0.338** | **0.390** | 0.390 | 0.412 | 0.354 | 0.391 | 0.384 | 0.407 | 0.383 | 0.407 | 0.393 | 0.413 | 0.942 | 0.684 | 0.387 | 0.407 | 0.559 | 0.515 | 0.450 | 0.459 |
| ECL | 96 | 0.137 | 0.236 | **0.135** | **0.229** | 0.178 | 0.264 | 0.153 | 0.244 | 0.148 | 0.240 | 0.173 | 0.275 | 0.219 | 0.314 | 0.181 | 0.270 | 0.197 | 0.282 | 0.201 | 0.317 |
| ECL | 192 | 0.155 | 0.252 | **0.155** | **0.249** | 0.189 | 0.280 | 0.166 | 0.256 | 0.162 | 0.253 | 0.195 | 0.288 | 0.231 | 0.322 | 0.188 | 0.274 | 0.196 | 0.285 | 0.222 | 0.334 |
| ECL | 336 | **0.174** | 0.272 | 0.175 | 0.271 | 0.209 | 0.296 | 0.184 | 0.275 | 0.178 | **0.269** | 0.206 | 0.300 | 0.246 | 0.337 | 0.204 | 0.293 | 0.209 | 0.301 | 0.231 | 0.338 |
| ECL | 720 | **0.195** | **0.293** | 0.225 | 0.316 | 0.242 | 0.321 | 0.226 | 0.313 | 0.225 | 0.317 | 0.231 | 0.335 | 0.280 | 0.363 | 0.246 | 0.324 | 0.245 | 0.333 | 0.254 | 0.361 |
| ECL | Avg | **0.165** | **0.263** | 0.173 | 0.266 | 0.204 | 0.290 | 0.182 | 0.272 | 0.178 | 0.270 | 0.201 | 0.300 | 0.244 | 0.334 | 0.205 | 0.290 | 0.212 | 0.300 | 0.227 | 0.338 |
| Traffic | 96 | **0.388** | 0.266 | **0.388** | **0.261** | 0.494 | 0.274 | 0.464 | 0.289 | 0.395 | 0.268 | 0.570 | 0.310 | 0.522 | 0.290 | 0.462 | 0.295 | 0.650 | 0.396 | 0.613 | 0.388 |
| Traffic | 192 | **0.408** | **0.270** | 0.421 | 0.279 | 0.487 | 0.323 | 0.477 | 0.292 | 0.417 | 0.276 | 0.577 | 0.321 | 0.530 | 0.293 | 0.466 | 0.296 | 0.598 | 0.370 | 0.616 | 0.382 |
| Traffic | 336 | **0.426** | **0.278** | 0.447 | 0.294 | 0.507 | 0.335 | 0.500 | 0.305 | 0.433 | 0.283 | 0.588 | 0.324 | 0.558 | 0.305 | 0.482 | 0.304 | 0.605 | 0.373 | 0.622 | 0.337 |
| Traffic | 720 | **0.449** | **0.292** | 0.511 | 0.330 | 0.548 | 0.359 | 0.548 | 0.313 | 0.467 | 0.302 | 0.597 | 0.337 | 0.589 | 0.328 | 0.514 | 0.322 | 0.645 | 0.394 | 0.660 | 0.408 |
| Traffic | Avg | **0.418** | **0.277** | 0.442 | 0.291 | 0.509 | 0.323 | 0.497 | 0.300 | 0.428 | 0.282 | 0.583 | 0.323 | 0.550 | 0.304 | 0.481 | 0.304 | 0.625 | 0.383 | 0.628 | 0.379 |
| Weather | 96 | 0.159 | **0.205** | 0.181 | 0.221 | 0.179 | 0.218 | 0.165 | 0.212 | 0.174 | 0.214 | 0.159 | 0.218 | **0.158** | 0.230 | 0.177 | 0.218 | 0.196 | 0.255 | 0.266 | 0.336 |
| Weather | 192 | **0.206** | **0.252** | 0.229 | 0.263 | 0.230 | 0.260 | 0.209 | 0.253 | 0.221 | 0.254 | 0.211 | 0.266 | **0.206** | 0.277 | 0.225 | 0.259 | 0.237 | 0.296 | 0.307 | 0.367 |
| Weather | 336 | **0.263** | **0.293** | 0.289 | 0.308 | 0.285 | 0.301 | 0.264 | **0.293** | 0.278 | 0.296 | 0.267 | 0.310 | 0.272 | 0.335 | 0.278 | 0.297 | 0.283 | 0.335 | 0.359 | 0.395 |
| Weather | 720 | 0.348 | 0.348 | 0.376 | 0.365 | 0.360 | 0.351 | **0.342** | **0.345** | 0.358 | 0.347 | 0.352 | 0.362 | 0.398 | 0.418 | 0.354 | 0.348 | 0.345 | 0.381 | 0.419 | 0.428 |
| Weather | Avg | **0.244** | **0.275** | 0.269 | 0.289 | 0.264 | 0.285 | 0.245 | 0.276 | 0.258 | 0.278 | 0.247 | 0.289 | 0.259 | 0.315 | 0.259 | 0.281 | 0.265 | 0.317 | 0.338 | 0.382 |
| Solar-Energy | 96 | 0.201 | 0.241 | 0.225 | 0.261 | **0.192** | 0.255 | 0.215 | 0.294 | 0.203 | **0.237** | 0.222 | 0.301 | 0.310 | 0.331 | 0.234 | 0.286 | 0.290 | 0.378 | 0.884 | 0.711 |
| Solar-Energy | 192 | 0.239 | 0.265 | 0.285 | 0.299 | **0.207** | 0.265 | 0.237 | 0.275 | 0.233 | **0.261** | 0.246 | 0.307 | 0.734 | 0.725 | 0.267 | 0.310 | 0.320 | 0.398 | 0.834 | 0.692 |
| Solar-Energy | 336 | 0.247 | 0.275 | 0.328 | 0.322 | **0.216** | **0.273** | 0.252 | 0.298 | 0.248 | **0.273** | 0.263 | 0.324 | 0.750 | 0.735 | 0.290 | 0.315 | 0.353 | 0.415 | 0.941 | 0.723 |
| Solar-Energy | 720 | 0.255 | 0.279 | 0.388 | 0.348 | **0.225** | **0.275** | 0.244 | 0.293 | 0.249 | **0.275** | 0.265 | 0.318 | 0.769 | 0.765 | 0.289 | 0.317 | 0.356 | 0.413 | 0.882 | 0.717 |
| Solar-Energy | Avg | 0.236 | 0.265 | 0.309 | 0.310 | **0.210** | 0.267 | 0.237 | 0.290 | 0.233 | **0.262** | 0.249 | 0.313 | 0.641 | 0.639 | 0.270 | 0.307 | 0.330 | 0.401 | 0.885 | 0.711 |

Table 9: Full short-term forecasting results (96 time-steps predicting {12, 24, 48, 96} time-steps): For short-term forecasting benchmark, we utilize 4 versions of well acknowledged PEMS datasets. We follow the experimental setup of (LIU et al., 2022; Liu et al., 2023). The average performance across all forecast horizons for each dataset are also reported.

| Models | | DELTAformer (Ours) | | Timer-XL (2025) | | SimpleTM (2025) | | TimeMixer (2024) | | iTransformer (2024) | | Crossformer (2023) | | PatchTST (2023) | | DLinear (2023) | | Autoformer (2021) | |
|---|---|---|---|---|---|---|---|---|---|---|---|---|---|---|---|---|---|---|---|
| Metrics | | MSE | MAE | MSE | MAE | MSE | MAE | MSE | MAE | MSE | MAE | MSE | MAE | MSE | MAE | MSE | MAE | MSE | MAE |
| PEMS03 | 12 | 0.062 | 0.165 | 0.059 | 0.160 | 0.076 | 0.183 | **0.060** | **0.163** | 0.071 | 0.174 | 0.090 | 0.203 | 0.099 | 0.216 | 0.122 | 0.243 | 0.272 | 0.385 |
| | 24 | 0.082 | 0.192 | 0.078 | **0.185** | 0.110 | 0.220 | **0.077** | 0.186 | 0.093 | 0.201 | 0.121 | 0.240 | 0.142 | 0.259 | 0.201 | 0.317 | 0.334 | 0.440 |
| | 48 | 0.121 | 0.235 | 0.136 | 0.247 | 0.223 | 0.311 | **0.109** | **0.221** | 0.125 | 0.236 | 0.202 | 0.317 | 0.211 | 0.319 | 0.333 | 0.425 | 1.032 | 0.782 |
| | 96 | 0.174 | 0.286 | 0.357 | 0.412 | 0.337 | 0.407 | 0.167 | **0.263** | **0.164** | 0.275 | 0.262 | 0.367 | 0.269 | 0.370 | 0.457 | 0.515 | 1.031 | 0.796 |
| | Avg | 0.110 | 0.220 | 0.158 | 0.251 | 0.186 | 0.280 | **0.103** | **0.208** | 0.113 | 0.221 | 0.169 | 0.281 | 0.180 | 0.291 | 0.278 | 0.375 | 0.667 | 0.601 |
| PEMS04 | 12 | 0.076 | 0.182 | 0.069 | 0.169 | 0.104 | 0.212 | **0.067** | **0.168** | 0.078 | 0.183 | 0.098 | 0.218 | 0.105 | 0.224 | 0.148 | 0.272 | 0.424 | 0.491 |
| | 24 | 0.097 | 0.201 | 0.085 | 0.191 | 0.163 | 0.277 | **0.073** | **0.184** | 0.095 | 0.205 | 0.131 | 0.256 | 0.153 | 0.275 | 0.224 | 0.340 | 0.459 | 0.509 |
| | 48 | 0.138 | 0.252 | 0.143 | 0.252 | 0.274 | 0.354 | **0.089** | **0.197** | 0.120 | 0.233 | 0.205 | 0.326 | 0.229 | 0.339 | 0.355 | 0.437 | 0.646 | 0.610 |
| | 96 | 0.191 | 0.308 | 0.451 | 0.454 | 0.494 | 0.492 | **0.102** | **0.216** | 0.150 | 0.262 | 0.402 | 0.457 | 0.291 | 0.389 | 0.452 | 0.504 | 0.912 | 0.748 |
| | Avg | 0.135 | 0.238 | 0.187 | 0.267 | 0.259 | 0.334 | **0.083** | **0.191** | 0.111 | 0.221 | 0.209 | 0.314 | 0.195 | 0.307 | 0.295 | 0.388 | 0.610 | 0.590 |
| PEMS07 | 12 | 0.058 | 0.157 | 0.069 | 0.174 | 0.086 | 0.194 | **0.055** | **0.149** | 0.067 | 0.165 | 0.094 | 0.200 | 0.095 | 0.207 | 0.115 | 0.242 | 0.199 | 0.336 |
| | 24 | 0.077 | 0.181 | 0.110 | 0.221 | 0.140 | 0.249 | **0.067** | **0.167** | 0.088 | 0.190 | 0.139 | 0.247 | 0.150 | 0.262 | 0.210 | 0.329 | 0.323 | 0.420 |
| | 48 | 0.107 | 0.217 | 0.244 | 0.325 | 0.250 | 0.344 | **0.087** | **0.196** | 0.110 | 0.215 | 0.311 | 0.369 | 0.253 | 0.340 | 0.398 | 0.458 | 0.390 | 0.470 |
| | 96 | 0.142 | 0.255 | 0.439 | 0.455 | 0.454 | 0.480 | **0.109** | **0.220** | 0.139 | 0.245 | 0.396 | 0.442 | 0.346 | 0.404 | 0.594 | 0.553 | 0.554 | 0.578 |
| | Avg | 0.096 | 0.203 | 0.215 | 0.293 | 0.233 | 0.317 | **0.0798** | **0.183** | 0.101 | 0.204 | 0.235 | 0.315 | 0.211 | 0.303 | 0.329 | 0.395 | 0.367 | 0.451 |
| PEMS08 | 12 | 0.070 | 0.173 | 0.062 | **0.157** | 0.075 | 0.177 | **0.061** | 0.161 | 0.079 | 0.182 | 0.165 | 0.214 | 0.168 | 0.232 | 0.154 | 0.276 | 0.436 | 0.485 |
| | 24 | 0.095 | 0.204 | 0.078 | **0.178** | 0.104 | 0.207 | **0.076** | 0.183 | 0.115 | 0.219 | 0.215 | 0.260 | 0.224 | 0.281 | 0.248 | 0.353 | 0.467 | 0.502 |
| | 48 | 0.142 | 0.254 | 0.135 | 0.237 | 0.167 | 0.266 | **0.104** | **0.220** | 0.186 | 0.235 | 0.315 | 0.355 | 0.321 | 0.354 | 0.440 | 0.470 | 0.966 | 0.733 |
| | 96 | 0.198 | 0.306 | 0.478 | 0.463 | 0.267 | 0.347 | **0.159** | 0.273 | 0.221 | **0.267** | 0.377 | 0.397 | 0.408 | 0.417 | 0.674 | 0.565 | 1.385 | 0.915 |
| | Avg | 0.126 | 0.234 | 0.188 | 0.258 | 0.153 | 0.249 | **0.100** | **0.209** | 0.150 | 0.226 | 0.268 | 0.307 | 0.280 | 0.321 | 0.379 | 0.416 | 0.814 | 0.659 |

dataset, we randomly select 3 batches and variables to plot, which we show above each chart. For Traffic and ECL, DELTAformer is able to learn complex patterns and show very robust predictions. However, DELTAformer does show that it struggles with spikes of particularly high magnitudes. For the Solar dataset, DELTAformer shows much more smooth and trivial patterns against the ground-truth signals that seem to be much more unpredictable and exhibit less repeatable patterns compared to Traffic and ECL.

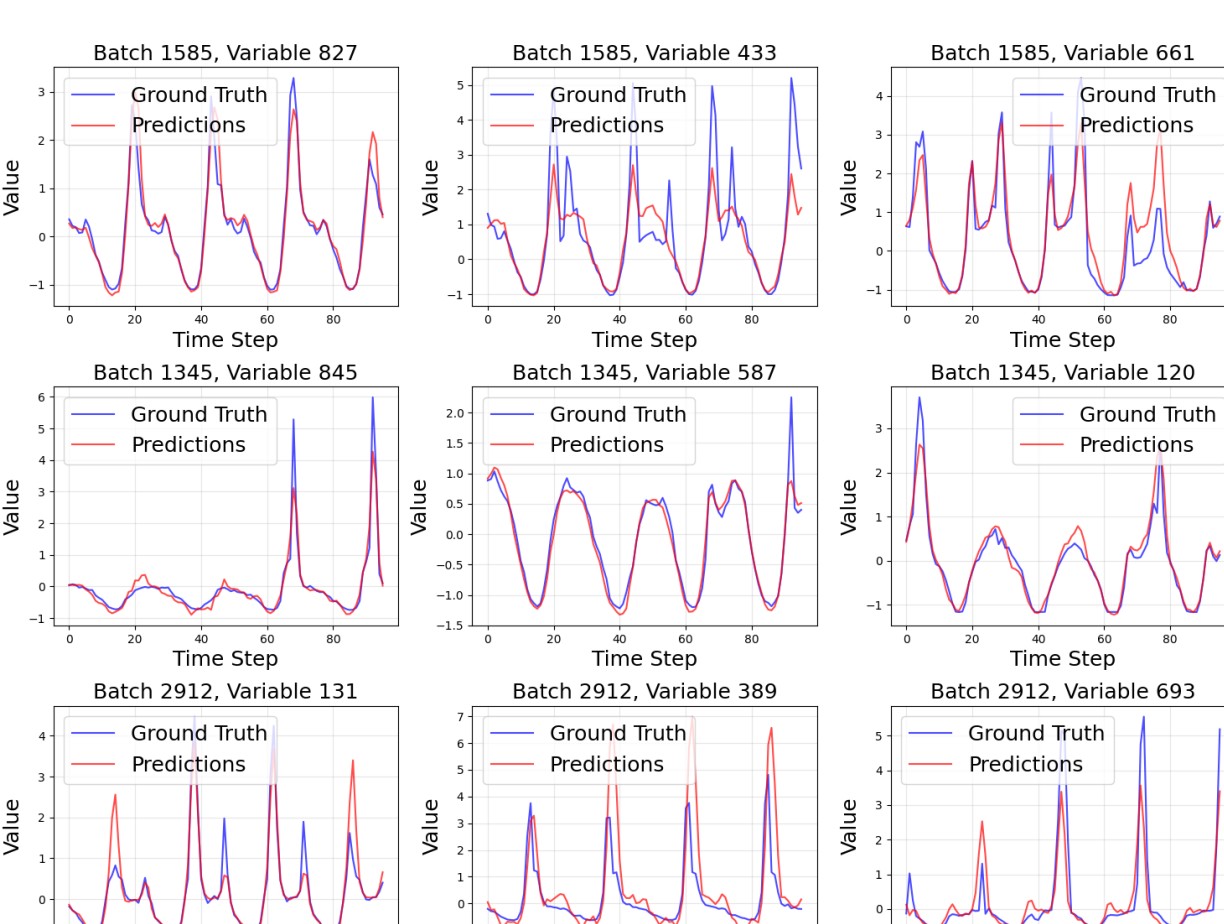

Figure 12

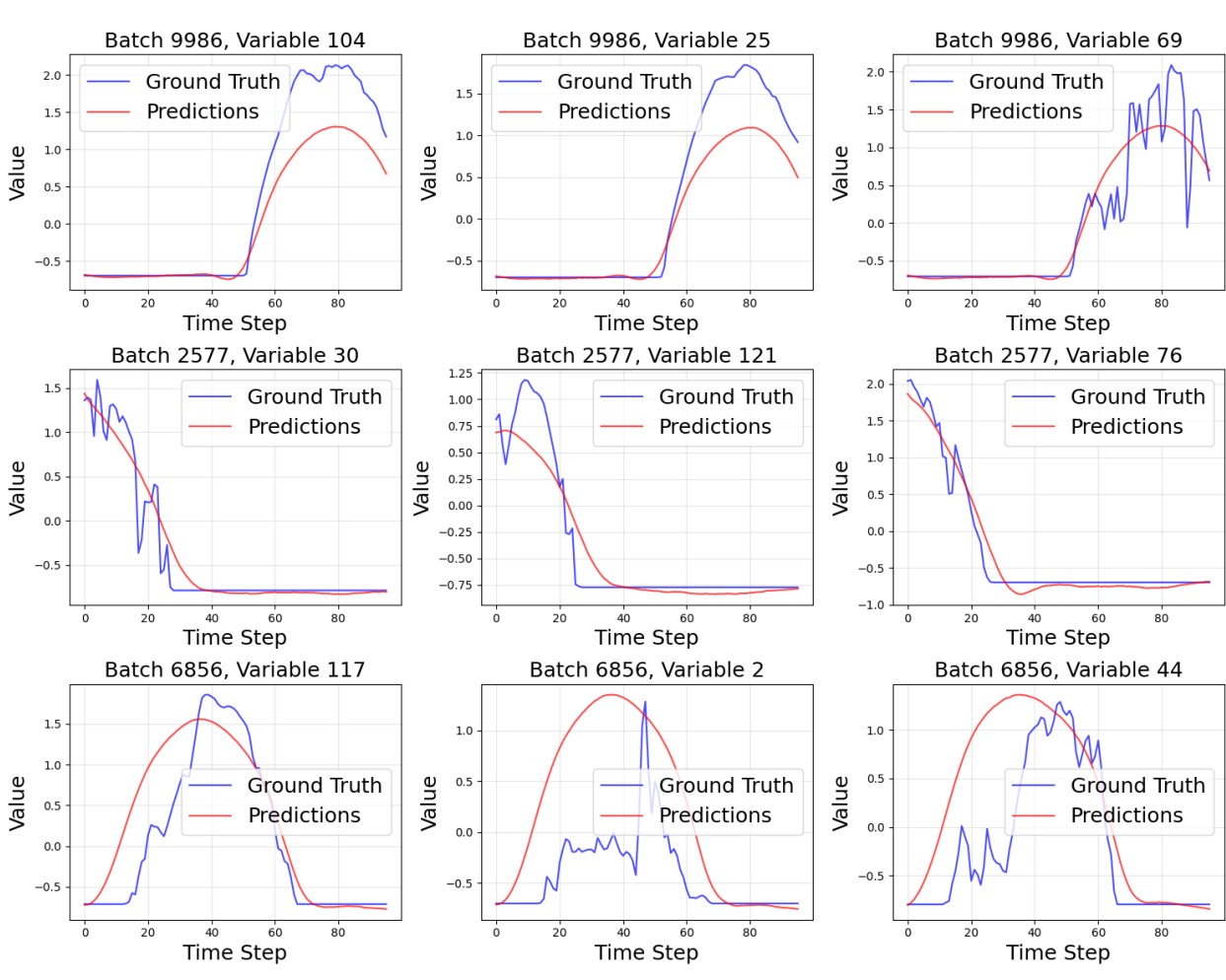

Figure 13

Figure 14

