# OpenReview forum: "Transformer Modeling for Both Scalability and Performance in Multivariate Time Series"
_TMLR — Rejected by TMLR_

### Review · Reviewer_bhBV · 2025-10-26

**Summary Of Contributions:**

The paper targets an important problem in multivariate time-series transformers: parameter counts that scale with the number of variables, which can lead to performance degradation. It proposes DELTAformer, which learns a bottleneck representation across variables, then models temporal dependencies with the help of attention in this representation, followed by an attention mechanism to decode values for each channel. The paper claims that DELTAformer achieves state-of-the-art performance on forecasting tasks and exhibits superior noise resilience.

**Audience:**

No

**Audience Explanation:**

The core idea is similar to router in Crossformer, and virtual nodes in GNNs. The paper positions its novelty as using the constrained representations themselves for temporal modeling relying on delegate-token self-attention rather than a computational shortcut. While this is a sensible distinction, it may also be viewed as a relatively minor extension of existing bottleneck paradigms. Limited novelty coupled with many missing baselines in the main result comparison raises the question if the proposed constrained representation deliver good performance. For these reasons, the paper may not be as interesting to the readers.

**Broader Impact Concerns:**

Broader Impact Statement is not included in the paper.

**Claims And Evidence:**

No

**Claims Explanation:**

* The paper claims that “DELTAformer achieves state-of-the-art performance on all long-term forecasting datasets.” However, I do not see the most recent convolutional [1] and frequency-domain [2,3] time series forecasting models included in the comparison, which raises concerns about the validity of the main claim of the paper.
* The main forecasting results considers 96 times-steps look back window length. Can the authors clarify why the full history is not fed to the models?

**References:**

[1] Luo, Donghao, and Xue Wang. "Moderntcn: A modern pure convolution structure for general time series analysis." The twelfth international conference on learning representations. 2024.

[2] Xu, Zhijian, Ailing Zeng, and Qiang Xu. "FITS: Modeling Time Series with $10 k $ Parameters." The Twelfth International Conference on Learning Representations.

[3] Wang, Hao, et al. "Fredf: Learning to forecast in the frequency domain." arXiv preprint arXiv:2402.02399 (2024).

**Requested Changes:**

Question: The Delegate Token Attention module that models dependencies in the time dimension *does not* perform causal attention. Why do you think pairwise comaprison across time is the correct approach here? (for example, why should future token have an influence on a historical token?)

Additions:
* Currently the work lacks many important baseline methods in the main results. Including at least one method from a class of methods will improve current work.
* Justify the 96-step context for all the models. An ablation on multiple lookback ranges will help understand if the proposed model can handle longer sequences.

Minor:
“Note: we use variable and channel inter-changeably in this work” can be formatted as a footnote.

---

### Review · Reviewer_pqEn · 2025-11-14

**Summary Of Contributions:**

This paper introduces the Deltaformer, a new attentional architecture for processing timeseries data, particularly well suited for situations where the number of data channels C far exceeds the number of time patches L/P. This model is quadratic in L/P, but linear in C. It accomplishes this by, for each time patch, tracking both tokens per-channel and a single “delegate” token. It performs attention to update these delegate tokens with all the per-channel tokens at the same timestep, and then does a full attention across all the delegate tokens across timesteps. It then updates the per-channel tokens using the delegates. This architecture achieves state-of-the-art performance on a range of benchmarks, while using less compute and memory; some experiments suggest this is because it attends more precisely to relevant signals, with the delegate mechanism forming a sort of regularization.

**Audience:**

Yes

**Audience Explanation:**

I believe so, though I am not familiar with the literature in this area.

**Broader Impact Concerns:**

I do not believe that there are any broader impact concerns related to this work.

**Claims And Evidence:**

Yes

**Claims Explanation:**

The main claims of this paper are well-substantiated by the results in Sections 4.2 and 4.1.

**Requested Changes:**

Section 2 needs to describe clearly that the setting being considered is one where L/P << C; this is not true in all timeseries domains; and as such a reader might be confused as to why C is considered the main variable to optimize performance against.

4.1 and 4.2 should be swapped in positions; the results in 4.2 are fairly impressive, but placing 4.1 first makes it seem to a reader reading the paper linearly like the Deltaformer is primarily useful at these secondary metrics. Section 4.1 becomes more interesting once you know that it is an explanation for better holistic performance.

Minor:

Equations in general: use \mathrm for softmax, funnel_in, LayerNorm, etc. Use \left( and \right) when appropriate.

Equation 1: what is d_M? It should be defined.

Right after equation 2: why would d_patch’ have to be a multiple of d_patch?

Equation 6: the LHS is missing the subscript [:, :, i]

---

### Review · Reviewer_Z5iB · 2025-11-18

**Summary Of Contributions:**

The paper proposes DELTAformer (Delegate Token Attention Transformer), a novel architecture for Multivariate Time Series (MTS) forecasting that simultaneously addresses two primary limitations of standard Transformer models: quadratic scalability with respect to the variable count ($C$) and performance degradation stemming from indiscriminate inter-variable mixing.

DELTAformer introduces a Delegate Token Attention mechanism which strategically constrains the flow of cross-variable information. This mechanism involves a three-stage funneling process: variables are aggregated into delegate tokens (DelTs) via Funnel-in Attention; unconstrained temporal dependencies are modeled among DelTs via Delegate Token Attention ($O((L/P)^2)$); and results are propagated back to the patches via Funnel-out Attention. This constraint acts as an implicit regularizer, forcing the model to be highly selective about the inter-variable information propagated, thereby enhancing signal detection in sparse MTS data.

The model achieves linear scalability $O((L/P)^2 + C(L/P))$ with respect to $C$, resolving the quadratic bottleneck of full and variate-only transformers. Empirically, DELTAformer achieves State-of-the-Art (SOTA) performance on all long-term forecasting benchmarks, demonstrating superior performance and robustness to noise and attention allocation efficiency compared to standard quadratic transformers.

**Additional Comments:**

The experimental setup, including the adoption of variate-wise patching and simple pre-processing techniques (RevIN), is appropriate for fair benchmarking against other modern Transformer baselines. The reported standard deviations confirm that the main forecasting results are stable and repeatable across runs. The comprehensive analysis comparing different funneling operations (Attention vs. MLP/Linear) effectively validates the architectural design choice, especially for high-dimensional data.

**Audience:**

Yes

**Audience Explanation:**

The paper addresses a crucial, well-recognized bottleneck in deep learning applied to time series—the quadratic scaling of Transformer models with variable count, which fundamentally limits their applicability in high-dimensional MTS domains. By providing a solution, DELTAformer, that simultaneously achieves linear complexity and SOTA performance, the findings are of significant interest to researchers focused on efficient model architecture design, time series forecasting, and applications in domains like smart city traffic flow modeling and energy forecasting. The demonstration that **strategically constraining representation capacity** can improve both efficiency and predictive accuracy is a valuable methodological contribution.

**Broader Impact Concerns:**

No broader impact concerns or ethical implications requiring a dedicated statement were discussed or raised in the provided sources.

**Claims And Evidence:**

Yes

**Claims Explanation:**

The claims regarding both scalability and performance are strongly supported by comprehensive empirical evidence.

1.  Scalability: DELTAformer's linear complexity $O((L/P)^2 + C(L/P))$ is verified through controlled experiments demonstrating a clear linear profile in memory usage against increasing variable count, contrasting sharply with the quadratic growth observed in full and variate-only transformers (Figure 6, 8, 9). Practical application confirms this: DELTAformer achieved 80% and 97% reductions in peak memory allocation against iTransformer and Timer-XL, respectively, on the high-dimensional Traffic dataset.
2.  Performance and Robustness: DELTAformer achieves SOTA performance across eight long-term forecasting benchmarks (e.g., 4.6% MSE reduction on ECL). Furthermore, dedicated analysis confirms the hypothesis regarding constrained mixing: DELTAformer consistently dedicates higher proportions of attention scores to relevant signals in key-retrieval analogue tasks (Figure 3) and shows superior resilience against artificially injected Gaussian noise across datasets like ECL, Solar, and Traffic (Figure 4).
3.  Architectural Validity: Ablation studies confirm that using the Attention mechanism for both Funnel-in and Funnel-out operations is crucial for robust MTS modeling, achieving up to 26% and 27% MSE improvements compared to MLP and linear layer alternatives, validating the core components of the constrained architecture.

**Requested Changes:**

First, strengthen the theoretical foundation: the current model design relies heavily on empirical findings and prior observations, and a rigorous theoretical basis is needed to better validate its claimed robustness. Second, investigate the suboptimal short-term forecasting performance—evidenced by DELTAformer’s underperformance against MLP models (e.g., TimeMixer) on PEMS datasets—to clarify how temporal distance modeling hinders the isolation of short-term interactions. Third, explore two specific limitations: (1) the model’s dependence on the assumption of sparse informative signals in multivariate time series (MTS) data (via testing on synthetic datasets with intentionally dense inter-variable information); (2) its inferior performance on low-dimensional data (e.g., ETTh1 with few variables) compared to other baselines.

---

> ### Author Response · Authors · 2025-11-20
> **Author response (part 1)**
>
> We thank the reviewer for the thorough and positive evaluation of our work. We aim to address reviewer comments to further strengthen and improve our work for the final version. This is part 1 of our response in 2 parts.
>
> ### On strengthening theoretical foundation
> The theoretical motivation of DELTAformer's constrained inter-variable architecture can be interpreted through the lens of the information bottleneck (IB) theory [1] and its application to deep learning [2]. At each time patch $i, i \in \{L/P\}$, DELTAformer passes all inter-variable information $M_i, M \in \mathbb{R}^{C \times d_{patch} \times (L/P)}$ to the temporal attention stack through a limited number of delegate tokens $D_i, D \in \mathbb{R}^{(L/p) \times d_{patch}}$ produced by the funnel-in attention. This means that for any downstream representation $H_i$ and prediction $Y$, the only path from $M_i$ to $H_i$ $\forall i \in \{L/P\}$ is through the delegate tokens. So $D \in \mathbb{R}^{(L/P) \times d_{patch}}$ form an explicit bottleneck in the sense of the IB method [1-2].
>
> The classical IB principle points to finding representation $Z$ that maximizes $I(Z;Y)$ while minimizing $I(Z;X)$, i.e. minimum amount of information from input data needed to get maximal performance on the output. In DELTAformer, delegate tokens can be interpreted as an explicit implementation of a learnable compression map $f_{\theta} = M_i \rightarrow Z_i$, which coupled with model optimization criteria of minimizing forecasting error and sparsity of informative signals in typical MTS data, potentially biasing the model toward discarding non-predictive inter-variable information (i.e. minimizing $I(Z;X)$) while learning a maximization strategy for mutual information $I(Z;Y)$.
>
> **Notable points to consider**: while our approach is in line with the IB view of deep representations, we point out that we do not explicitly optimize for the IB Lagrangian. Instead, we find that implicitly encouraging IB-style constraints through our bottleneck design is enough to attain robust performance. Indeed, our empirical findings are in line with IB-style designs (higher attention concentration, greater noise resilience, etc.) in addition to outperforming many full transformers. As such, our paper is positioned to be an empirically-driven paper, with theoretical inspirations acting as architectural motivation in a domain-appropriate manner, but not a theoretical study.
>
> [1] Tishby, Naftali, Fernando C. Pereira, and William Bialek. "The information bottleneck method." _arXiv preprint physics/0004057_ (2000).
> [2] Tishby, Naftali, and Noga Zaslavsky. "Deep learning and the information bottleneck principle." _2015 ieee information theory workshop (itw)_. Ieee, 2015.
>
> ### Addressing model dependence on assumption of sparse signals
> Regarding the model's dependence on the assumption of sparse informative signals, we aim to provide more information here. We generally found it difficult to construct synthetic experimental setups that contained dense inter-variable signals while also being uncompressible. Dense signals that can effectively be compressed were not able to test DELTAformer's architecture effectively (for exmaple sinusoidal signals or any other periodic signals can effectively be compressed and therefore DELTAformer does not struggle against them). On the other hand, synthetic signals that were more random and unpredictable were very difficult to train on, not just on DELTAformer but on other transformer models as well, leading to very high overfitting and arbitrarily bad performance at inference time. Overall, we found this experimental setup to be particularly challenging, struggling to find the right balance in data that were learnable but challenging for DELTAformer. We still agree with the reviewer that DELTAformer may face challenges when encountering MTS data with particularly dense signals. Relating this to the IB principle from above response, the irreversibility of information as laid out by the IB principle (If $X \rightarrow Y \rightarrow Z$ forms a Markov chain, then $I(X;Z) \leq I(X;Y)$) strongly implies that dense information may require bigger delegate tokens. We believe that our choice to make the delegate token size into a hyperparameter (delegate token dimensionality is de-coupled from regular patch token size and can be tuned) may be useful in this regime (increasing delegate token size decreases the bottleneck). We still note that we haven't found this to be necessary and that most practical MTS systems do display sparse information patterns.

---

> ### Author Response · Authors · 2025-11-20
> **Author response (part 2)**
>
> ### On suboptimal performance on short-term/low-dimensional systems
> As for the suboptimal performance on short-term forecasting, we see a general trend of suboptimal performance from transformer architectures overall. This may be a consequence of transformers' large receptive field, which are advantageous when long-term trends need to be identified but could serve to accumulate undue noise when the objective is short-term pattern identification. In this regime, DELTAformer still generally outperforms full transformers and remains relatively competitive to models like TimeMixer, possibly due to its enhanced noise-resilience. This also speaks to the lack of clear foundation models in this space, as MTS data incur too much diversity (sometimes even within short-term or long-term forecasts) and models optimized for one regime often struggle in another. We believe DELTAformer makes a solid attempt at bridging this gap, using the sparsity of informative inter-variable signals as the unifying challenge to be solved in multivariate time series and achieving SOTA performance on many long-term benchmarks while still getting competitive performance on short-term benchmarks, albeit showing weaker performance compared to models like TimeMixer.
>
>  On weaker performance on low-dimensional datasets like ETT variants, DELTAformer still gets SOTA performance here overall but the reviewer raises an important question worth investigating. We believe our above assessment about models excelling in one regime of MTS struggling in another also holds in this case. On low-dimensional datasets, simpler models like channel-independent SimpleTM and MLP based TimeMixer tend to perform better than transformers (up to 7.3\% improvement on ETT against DELTAformer). On the other hand, on high-dimensional datasets like Traffic (N=862) and ECL (N=321), these simpler models are substantially worse than transformer models. DELTAformer shows a 15.9\% and 9.3\% improvement compared to TimeMixer on Traffic and ECL, respectively. DELTAformer's dominant SOTA performance on high-dimensional datasets coupled with its overall SOTA results on combined ETT variants (though DELTAformer does show inferior performance on some ETT variants) show that DELTAformer presents a flexible and one-size-fits-many solution in a regime that is notoriously diverse and challenging.

---

### Decision · Action_Editor_8F5F · 2026-01-13

**Recommendation:** Reject

**Additional Comments:**

The majority of the reviewers leaned towards rejecting the paper, and after reading the paper and reviews, I concur with their decision.

Final decision: Reject.
However, authors, if they like, may consider  submitting a major revision, addressing all of the concerns raised by the reviewers, at a later time.

**Audience:**

No

**Audience Explanation:**

The paper positions delegate tokens as constrained data representations and suggests that this design provides a meaningful methodological contribution. However, as described by the authors themselves, several aspects of the approach closely resemble existing transformer mechanisms and previously studied forms of representational bottlenecks. The paper does not clearly articulate a sufficiently distinct insight, capability, or practical implication that would make the findings broadly informative or useful to the TMLR audience.

In particular, it remains unclear how the proposed formulation advances understanding beyond existing methods that employ structured token interactions or bottleneck-style representations. As a result, even if the method is technically sound, the paper does not clearly identify a takeaway that would justify strong interest from a segment of the TMLR readership.

**Claims And Evidence:**

No

**Claims Explanation:**

Several of the paper’s central claims, particularly those related to robustness, are not supported by sufficiently rigorous evidence. Beyond empirical observations and references to prior work, a clearer theoretical justification is needed to substantiate these claims.

Moreover, the experimental results reveal suboptimal performance in short-term forecasting tasks, as well as inferior performance on low-dimensional data when compared to existing baselines. These outcomes are not adequately discussed or explained in the paper, leaving important questions about the scope and limitations of the method unanswered.

Finally, the absence of several important baseline comparisons further weakens the empirical validation. Without these baselines, it is difficult to determine whether the reported results substantiate the paper’s implied claims of effectiveness relative to existing approaches.

**Resubmission Of Major Revision:**

The authors may consider submitting a major revision at a later time.